

# Investigation of Gravity Waves using Measurements from a Sodium Temperature/Wind Lidar Operated in Multi-Direction Mode

Bing Cao[1] and Alan Z. Liu[2]

[1]Institute of Geophysics and Planetary Physics, Scripps Institution of Oceanography, University of California at San Diego, La Jolla, California
[2]Center for Space and Atmospheric Research, Department of Physical Sciences, Embry-Riddle Aeronautical University, Daytona Beach, Florida

**Correspondence:** Bing Cao (bic020@ucsd.edu)

**Abstract.** A narrow-band sodium lidar provides high temporal and vertical resolution observations of sodium density, atmospheric temperature, and wind that facilitate the investigation of atmospheric waves in the mesosphere and lower thermosphere (80–105 km). In order to retrieve full vector winds, such a lidar is usually configured in a multi-direction observing mode, with laser beams pointing to the zenith and several off-zenith directions. Gravity wave events were observed by such a lidar system from 06:30 to 11:00 UTC on 14 January 2002 at Maui, Hawaii (20.7° N, 156.3° W). A novel method based on cross-spectrum was proposed to derive the horizontal wave information from the phase shifts among measurements in different directions. At least two wave packets were identified using this method, one with a period of ∼1.6 hr and a horizontal wavelength of ∼438 km and propagating toward the southwest, the other one with a ∼3.2 hr period and ∼975 km wavelength and propagating toward the northwest. The background atmosphere states were also fully measured, and all intrinsic wave properties of the wave packets were derived. Dispersion and polarization relations were used to diagnose wave propagation and dissipation. It was revealed that both wave packets propagate through multiple thin evanescent layers and are partially reflected but still get a good portion of energy to penetrate higher altitudes. A sensitivity study demonstrates the capability of this method in detecting medium-scale and medium-frequency gravity waves. With continuous and high-quality measurements from similar lidar systems worldwide, this method can be utilized to detect and study the characteristics of gravity waves of specific spatio-temporal scales.

## 1 Introduction

Atmospheric gravity waves are generated when air parcels are perturbed vertically, and gravity/buoyancy acts as the restoring force. They can propagate vertically up to the thermosphere and horizontally over a considerable distance (up to several thousand kilometers). The most common wave sources include convection, orography, and fronts, etc. (Fritts and Alexander, 2003, and references therein). The momentum and energy transported by gravity waves dramatically impact the general circulation and thermal structure of the middle and upper atmosphere. Phenomena and processes include, but are not limited to, the cold summer mesopause (Holton, 1982; Siskind et al., 2012), the quasi-biennial oscillation (QBO) in tropical lower stratosphere (Ern et al., 2014) and the semiannual oscillation (SAO) in the upper stratosphere/lower mesosphere (Ern et al., 2015), instability





and turbulent mixing in the atmosphere (Fritts, 1984; Fritts et al., 2013), and irregularities and traveling ionospheric distur-

bances (Fritts and Lund, 2011; Liu and Vadas, 2013), are all related to the gravity waves. Therefore, understanding gravity wave generation, propagation, and breaking has significant impacts on weather and climate applications.

Theoretically, gravity waves are governed by the fluid Euler equations for a set of fundamental variables including pressure $p$, density $\rho$, temperature $T$, and zonal, meridional and vertical winds ($u$, $v$, $w$). Many theoretical and observational studies of gravity waves are based on the linear wave theory, and it is commonly used to describe the propagation and dissipation

characteristics of gravity waves. The linearization of the Euler equations is implemented under different assumptions regarding wave and background atmosphere properties. Except for waves with very large horizontal scales, the effect of Earth rotation is often ignored. Zhou and Morton (2007) derived the Euler equations for a compressible atmosphere with altitude-varying background temperature and wind. Taylor (1931) and Goldstein (1931) derived the 2-D Euler equations with the Boussinesq approximation in a continuous shear flow without temperature variations. These specific Euler equations are referred to as

Taylor-Goldstein equations (Nappo, 2012). Fritts and Alexander (2003) derived the Euler equations without wind shear but considered the Coriolis effect. Linearized wave solutions require that the vertical wavenumber $m$ is independent of altitude. Strict independence is not likely since background temperature and wind vary with altitude. If the variations are relatively slow within the range of vertical wavelength, the Wentzel-Kramer-Brillouin (WKB) approximation is applied. A monochromatic gravity wave can be assumed to be a traveling plane wave and represented by a sinusoidal function in the form of:

$$W(x,y,z,t) = A \cdot \exp\left[i(kx + ly + mz - \omega t + \phi) + \frac{z}{2H_s}\right], \tag{1}$$

of which $x$, $y$, and $z$ are zonal, meridional and vertical distances in a local Cartesian coordinate, and $k$, $l$, and $m$ are zonal, meridional and vertical wavenumber. $\omega$ and $\phi$ are the observed (Eulerian) angular wave frequency and initial phase, $H_s$ is the atmospheric density scale height, and $1/(2H_s)$ corresponds to the wave amplitude growth rate with increasing altitude for upward propagating and non-dissipating gravity waves.

Many remote-sensing and in-situ techniques have been developed to observe atmospheric gravity waves and their influences

in past decades. There is an inherent 'observation filter' (Alexander, 1998; Gardner and Taylor, 1998; Alexander et al., 2010) in any of these observation instruments such that they can only measure part of the gravity wave spectrum. Single-site ground-based techniques like lidar (Hu et al., 2002; Li et al., 2007; Lu et al., 2009; Cai et al., 2014; Chen et al., 2016), radar (Nastrom and Eaton, 2006; Fritts et al., 2010; Liu et al., 2013), and space-based limb sounding satellites (Jiang et al., 2005; Alexander et al., 2009) are limited to providing vertical profiles and can only resolve vertical structures of the wave field. Other techniques

like nadir sounding satellites (Gong et al., 2012; Alexander and Grimsdell, 2013; Hoffmann et al., 2014) and airglow imaging (Taylor, 1997; Espy et al., 2006; Li et al., 2011; Fritts et al., 2014; Cao and Liu, 2022) can only retrieve the horizontal structures over a certain area. In some cases, the unobserved horizontal or vertical information can be estimated by indirect methods based on the polarization and dispersion relationships (Hu et al., 2002; Lu et al., 2015). For reliable estimates of intrinsic wave parameters and characterization of the dissipation process, it is necessary to observe gravity waves fully in both

horizontal and vertical directions, i.e., in 3-D space. Measurements from multiple complementary instruments, such as co-located lidar and airglow imager, provide good opportunities to resolve gravity waves in both vertical and horizontal directions



(Bossert et al., 2014; Lu et al., 2015; Cao et al., 2016). More space-borne limb-sounding observations were also analyzed with special techniques to resolve the gravity wave as fully as possible. Two-dimensional (2-D) limb-sounding measurements along a single satellite track, such as temperature variations from High-Resolution Dynamics Limb Sounder (HIRDLS), were

used to estimate the horizontal wavelength and momentum flux for individual wave events (Alexander et al., 2008). However, there were considerable uncertainties in the derived wave properties because the 2-D sampling measures only the apparent wavelength, which is likely longer than the actual ones. This was also illustrated by combining rarely occurring colocations of two HIRDLS profiles and one COSMIC Radio Occultation (RO) profile (Alexander et al., 2015), which showed the 2-D method overestimated the occurrence of long horizontal wavelength waves relative to the 3-D method and underestimated momentum

flux. Multiple methods have been proposed to resolve the intrinsic horizontal propagation and momentum flux assuming a dominant wave mode within a spatial (latitude and longitude) and temporal window large enough that it is coherent across more than three RO profiles (Wang and Alexander, 2010; Faber et al., 2013). Several studies have analyzed the horizontal wave information from closely spaced profiles in serendipitous geometries, for example, from COSMIC satellites just after launch and before they have separated into their final orbits. Alexander et al. (2018) used four spatially and temporally close

RO profiles to derive intrinsic wave propagation and wavelength.

This study provides a prospective solution to partly address the 'observation filter' effect of the narrow-band sodium lidars. The measurement profiles retrieved from these lidar systems were typically used to observe the temporal and vertical variations of the waves. We proposed a novel method using cross-spectrum to retrieve the often neglected horizontal wave information from lidar measurements. A case study of gravity wave events being fully captured in 3-D space is presented

based on the proposed method. Temperature and wind measurements on the night of 14 January 2002 from a lidar deployed in the Maui/Mesosphere and Lower Thermosphere (Maui/MALT) campaign are used to retrieve the wave information. The paper is organized as follows: Section 2 describes the instrumentation and methodology. Section 3 presents the observational results and diagnostic analysis. Section 4 presents a sensitivity study to demonstrate the capability of this method. Finally, the discussions and summary are presented in Section 5. Extra figures are included in the Appendix as supporting information.

## 2    Instrumentation and Methodology

A narrow-band sodium lidar measures the atmospheric temperature and wind in the mesopause region (80–105 km) based on the thermal broadening and Doppler shift of atomic spectral lines of the sodium atoms. The sodium lidar transmits pulsed laser tuned to the sodium D2a line at 589.158 nm into the sky and the fluorescence scattered photons are collected by optical telescopes. The temperature and line-of-sight (LOS) wind are retrieved based on the well-known shape of the Na atomic

spectrum using a three-frequency technique (She and Yu, 1994; Krueger et al., 2015). In order to measure the full 3-D wind vectors, the laser beam is configured to point to multiple directions, at zenith and off-zenith at several cardinal directions. A sodium lidar system operated by the University of Illinois at Urbana-Champaign (UIUC) was deployed at Air Force Maui Optical Station (AMOS) at Maui, HI ($20.7°$ N, $156.4°$ W) from Jan 2002 to Jun 2007 (Liu, 2023). The laser transmitter was fixed but the laser beam was directed by multiple mirrors to point to five directions: zenith ($Z$), $30°$ off zenith to the north



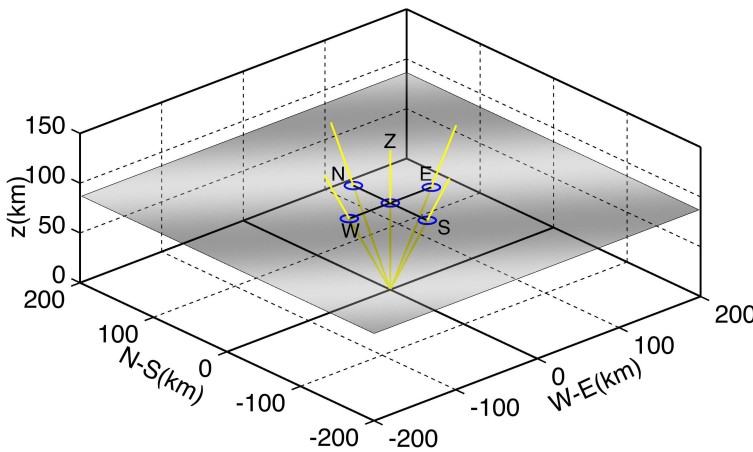

**Figure 1.** Diagram of a lidar operated in 5-direction observation mode. Yellow lines represent the laser beams pointed in five different directions, which are the outcome of the rotation of one laser beam. The laser beam's off-zenith angle is 30°. A plane wave is shown in greyscale at 90 km altitude with 200 km horizontal wavelength and the wavefront is oriented at 60° clockwise from north.

($N$), south ($S$), east ($E$), and west ($W$). The laser beam has an average output power of 1 W at 30 pps (pulse per second). A steerable astronomical telescope of 3.67 m diameter was coupled with the laser beam. The return photons were collected by the telescope pointing in the same direction as the laser beam. Figure 1 is a diagram showing the orientation of laser beams in five directions. With a 30° off-zenith angle, there is a ∼50 km separation distance between off-zenith and the zenith directions at 90 km altitude. The laser beam was directed to rotate in $ZNEZSW$ sequence and the photon integration time is 1.5 min at

each direction with a 1.7 min cadence (extra ∼0.2 min for telescope steering). The resulting measurement intervals at zenith and any off-zenith directions are 5.1 min and 10.2 min; however, some irregularities exist because of mechanic issues. In order to accommodate the data processing of filtering and spectral analysis, the raw data of all directions was interpolated into the regular 6-min interval. The spatial resolution is 500 m along the laser beam in all directions. The temperature and winds from off-zenith directions are interpolated to the same altitude grids as the $Z$ direction with a uniform 500-m resolution. The lidar

measured temperature and wind accuracies largely depend on the sodium atom density; thus they vary with altitudes. With the temporal and spatial resolution in this study, the accuracies are ∼2 K for temperature and ∼4 m s$^{-1}$ for horizontal winds near the peak sodium density altitudes (Li et al., 2012; Krueger et al., 2015). The most reliable measurements are mainly within the altitude range 85–105 km. The mean sodium atom density and the altitude-varying accuracies for temperature, horizontal and vertical winds are attached in the Appendix (Figure A1).





The relationship between LOS winds ($V_E$, $V_W$, $V_N$, $V_S$, $V_Z$) and zonal, meridional and vertical winds ($u_x$, $v_y$, $w_z$) at different directions ($x = E$, $W$, $y = N$, $S$, $z = E$, $W$, $N$, $S$, $Z$) are described by (Gardner and Liu, 2007, equation A1)

$$V_E = u_E \sin\alpha + w_E \cos\alpha$$
$$V_W = -u_W \sin\alpha + w_W \cos\alpha$$
$$V_N = v_N \sin\alpha + w_N \cos\alpha \qquad (2)$$
$$V_S = -v_S \sin\alpha + w_S \cos\alpha$$
$$V_Z = w_Z,$$

where $\alpha$ is the off-zenith angle of laser beams. Under the assumption of homogeneity among different directions and vertical winds much smaller than the horizontal winds, the vertical winds at off-zenith directions are assumed to be the same as zenith direction. Therefore, the zonal, meridional and vertical winds are derived from LOS winds as follows:

$$u_E = (V_E - V_Z \cos\alpha)/\sin\alpha$$
$$u_W = -(V_W - V_Z \cos\alpha)/\sin\alpha$$
$$v_N = (V_N - V_Z \cos\alpha)/\sin\alpha \qquad (3)$$
$$v_S = -(V_S - V_Z \cos\alpha)/\sin\alpha$$
$$w_Z = V_Z.$$

Note that the derived zonal wind ($u_E$ and $u_W$), meridional wind ($v_N$ and $v_S$), and vertical wind ($w_Z$) are available in different directions and time steps. The scalar temperature measurements are available in all five directions. When temperature and wind measurements from this type of lidar were analyzed, the separations of laser beams among different directions were generally ignored under the assumption of spatial homogeneity among all laser beams. If a wave defined by equation (1) propagates through the five laser beams from a specific direction, the laser beams will probe different phases of the wave, as shown in

Figure 1. Therefore, phase differences can be determined from measurements in different directions and further used to derive horizontal wave information.

    The cross-spectral method was used to retrieve the phase shift among measurements of different directions. Given two time series of the same frequency $\omega$ but with different phases,

$$y_1 = A_1 \cdot \sin(\omega t + \phi_1)$$
$$y_2 = A_2 \cdot \sin(\omega t + \phi_2), \qquad (4)$$

the corresponding spectra are $Y_1(\omega) = FFT[y_1(t)]$ and $Y_2(\omega) = FFT[y_2(t)]$, and the cross-spectral between the two time

series is $I_{12}(\omega) = Y_1(\omega) \cdot Y_2^{\star}(\omega)$. The argument of the complex cross-spectrum is the phase difference between two waves ($\text{Arg}[I_{12}(\omega)] = \phi_1 - \phi_2$). The cross-spectral method treating the wave packets as quasi-monochromatic might not reflect the realistic dispersive waves but preserve the main characteristics. In the wave parameters estimation, there might exist ambiguity in the phase differences. Therefore, the wave horizontal wavelength must be larger than the laser beam separation which yields



a small phase difference for this method to be effective. Furthermore, the validity of derived wave speed could be used to
eliminate some unphysical results. There is also ambiguity in distinguishing the wave propagation direction. In this study,
measurements from the third available direction were utilized to resolve the ambiguity. Considering the uncertainties of the
measurements, the phase differences could not be too small to be distinguished. A sensitivity study was done to verify the
effectiveness of the method in resolving waves of different scales and periods.

This type of sodium lidar was usually operated at night, with a typical duration of no longer than 10 hours. The resulting
resolution ($1/10$ $\mathrm{hr}^{-1}$) in the spectral domain is relatively coarse. There exist possibilities of spectral leakage and the true
peaks falling between two adjacent integral spectral points. We improved the identification of the spectral peaks by applying
a nonlinear fitting of a parabolic function on the magnitudes of three integer spectral points close to the apparent peak. With
the refined method, it is hoped to acquire the non-integer spectral peaks closer to the actual values and estimate the wave
amplitudes $A$, periods $\tau$ ($T$ is reserved to abbreviate 'temperature'), and phase shifts $\phi$ more accurately. Given the off-zenith
angle $\alpha$, the spatial separation between zenith and any off-zenith directions can be calculated as $\Delta = h \cdot \tan(\alpha)$ at altitude $h$.
Once the phase differences in zonal and meridional directions ($\phi_x$ and $\phi_y$) are determined by the cross-spectral methods, the
horizontal wavenumbers in the zonal and meridional directions are derived from

$$
\begin{aligned}
k &= \frac{2\pi}{\Delta \cdot \phi_x} \\
l &= \frac{2\pi}{\Delta \cdot \phi_y}.
\end{aligned}
\tag{5}
$$

Then, the full set of horizontal wave parameters of wavelength $\lambda_H$, propagation azimuth angle $\theta$ and observed (ground-based)
phase speed $c_H$ can be calculated as

$$
\begin{aligned}
\lambda_H &= \frac{|k \cdot l|}{\sqrt{k^2 + l^2}} \\
\theta &= \arctan\left(\frac{l}{k}\right) \\
c_H &= \frac{\lambda_H}{\tau}.
\end{aligned}
\tag{6}
$$

## 3 Observational Results

### 3.1 Temperature/Wind Perturbations and Background States

On the night of January 14, 2002, the lidar was operated in 5-direction mode from around 6:30 to 11:00 UT. Sodium density,
temperature, and winds were continuously observed within the period from all five directions. The detrended temperature
measurements in different directions are shown in Figure 2. Abundant wave components of various periods are identified from
measurements of all directions, and distinct downward phase progression is seen in the perturbations, which implies an upward
wave propagation. The amplitudes of temperature perturbations reach $\pm 10$ K, and there is a layered structure in the vertical
direction. A strong peak is found at around 90 km altitude from 09:00 UT onwards. The wave patterns of the perturbations
in different directions are very similar, so they are likely the same wave packets spreading a larger area and captured by the



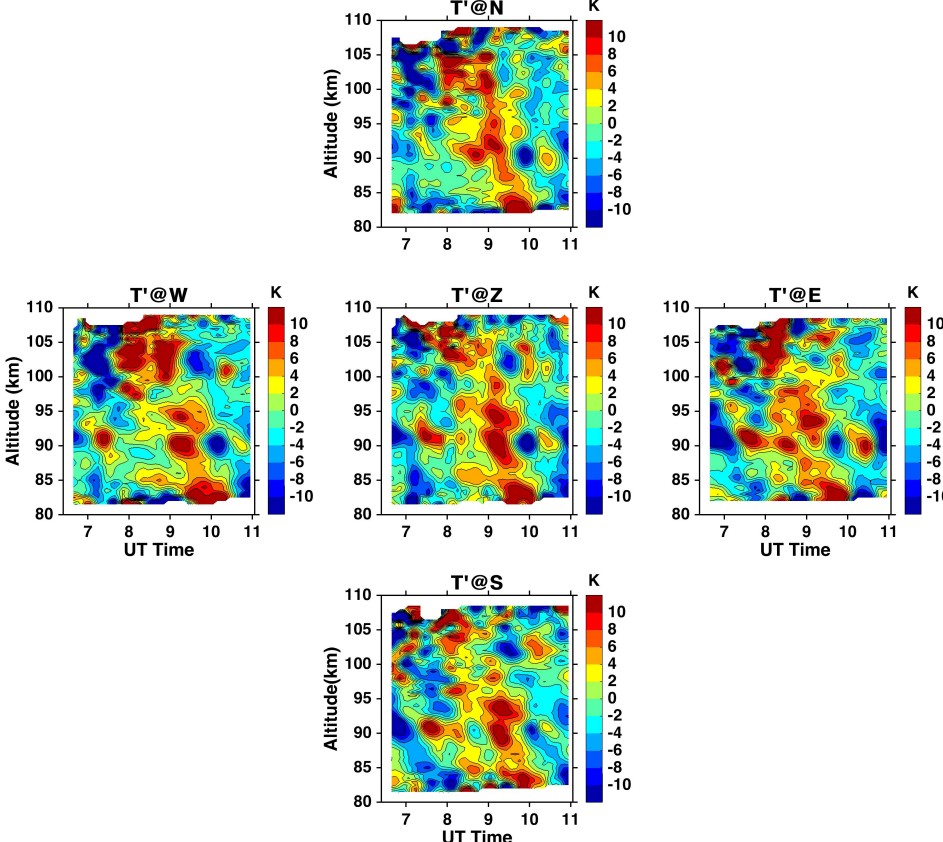

**Figure 2.** Detrended temperature perturbation in five different directions. The y-axis of non-zenith directions is the true altitude corrected from the slant distance along the laser beam.

laser beams in different directions. Closely inspecting the wave pattern (crests and troughs) in different directions, some shifts
in time could be noticed, which are the results of the spatial separation of laser beams in different directions. The detrended perturbations of different wind components are shown in Figure 3, similar wave patterns with a downward phase progression can be identified in zonal and meridional winds, with an amplitude of up to $\pm 20 \, \mathrm{m \, s^{-1}}$. The wave pattern is still clear in the vertical wind perturbation, with an amplitude $\pm 2 \, \mathrm{m \, s^{-1}}$. However, the downward phase progression is less evident. This is likely due to the magnitude of perturbation being equal to or less than the uncertainty of vertical winds. For both temperature
and winds, the measurements at the top and bottom sides are associated with larger uncertainties and should be interpreted with caution.

To identify the dominant wave modes from the temperature and winds measurements, the frequency spectra were calculated from detrended temperature and wind perturbations at all altitudes. Figure 4 shows the spectra of temperature perturbation in five directions, which all show a similar pattern. The average spectrum of all five directions is shown in the upper right corner.



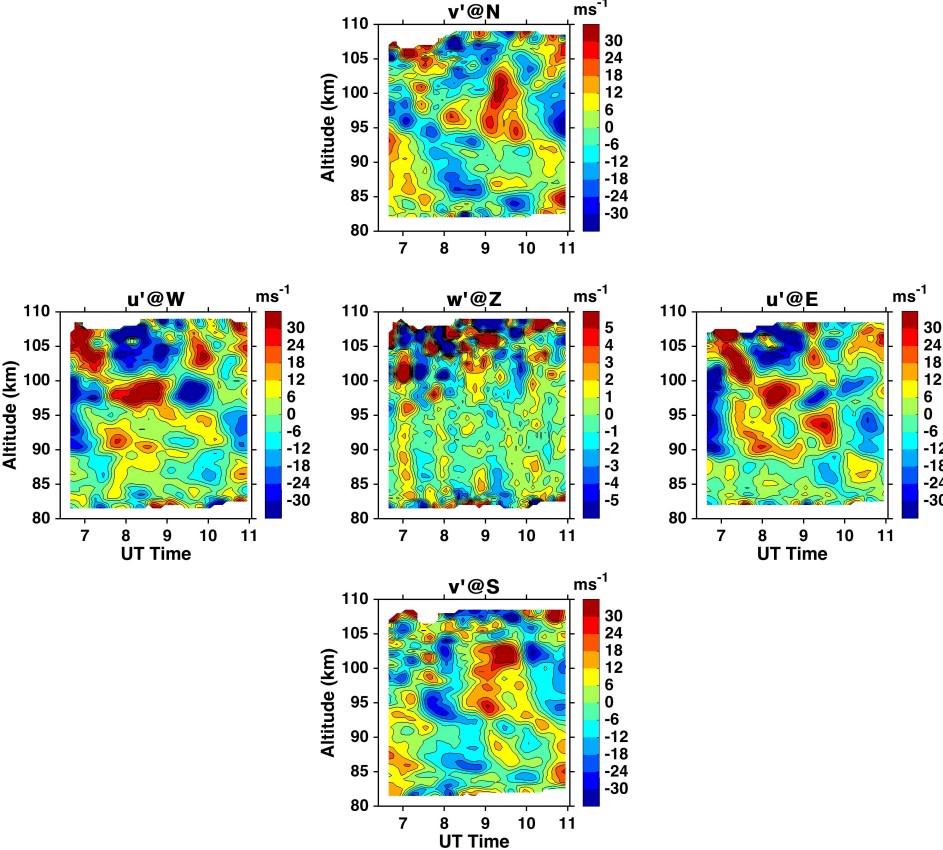

**Figure 3.** Detrended wind measurements in five different directions. Note that it is the zonal wind at $W$ and $E$, meridional wind at $S$ and $N$, and vertical wind at $Z$. The color scales for horizontal winds (zonal and meridional) and vertical wind are different.

The identified spectral peaks at each altitude using the fitting method were also denoted on the average spectrum. There might be one to several peaks at each altitude. Overall, there exist two prominent peaks; one has a period of about 3.2-hr and the other one about 1.6 hr. The 3.2-hr period component is persistent along with the whole altitude range, and the 1.6-hr peak exists mostly below 90 km and reaches a maximum at 90 km. The spectra of wind perturbation are shown in the Appendix (Figure A2). The spectral peak around 3.2-hr is also dominant in the horizontal (zonal and meridional) winds. However, the

1.6-hr one is less evident. This is likely due to the spectral leakage of the 3.2-hr wave component with much larger magnitudes which overwhelms the 1.6-hr period one. The two dominant wave components need to be separated from the mean background states, variations with longer periods and from each other for further cross-spectral analysis. Cut-off periods/frequencies are determined for desired digital filters based on the mean spectra of temperature and wind perturbations. Two spectral peaks are quite close in the frequency domain, so we used Chebyshev type II filters with flat passband and steep transition to stopband.

To filter out the 1.6-hr wave component, the cut-off period of a high-pass filter is selected as 2.2-hr ($0.46 \ \mathrm{hr}^{-1}$). To separate the background state from two wave components, the cut-off period of a low-pass filter is selected to be 6-hr ($0.17 \ \mathrm{hr}^{-1}$).



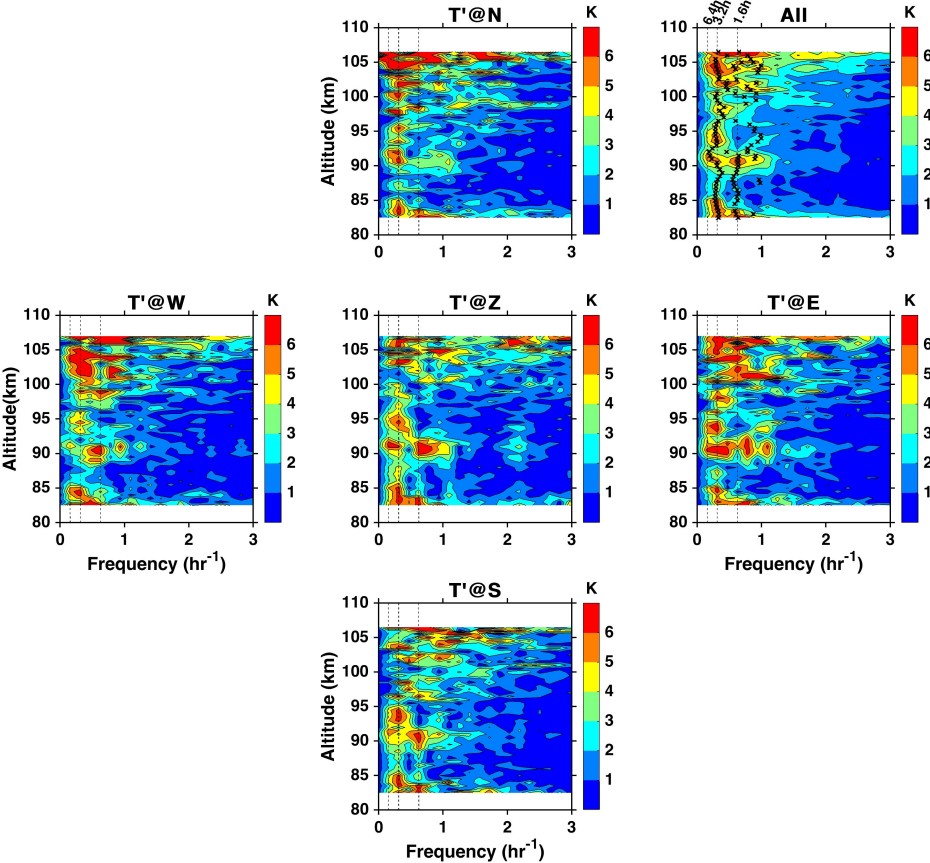

**Figure 4.** Spectra of temperature perturbation in all five directions, and the average of all five directions is shown in the upper-right corner, with black crosses marking the peaks at each altitude. See text about the method of determining those peaks. The vertical dashed lines denote the periods of 6.4-hr, 3.2-hr, and 1.6-hr.

To fully understand the propagation condition of waves, the background atmosphere states were analyzed. Figures 5(a)–5(c) show the background temperature $T_0$, zonal wind $u_0$ and meridional wind $v_0$ retrieved by low-pass filtering as defined above. The background atmosphere states show clear modulation of tides, as shown by a slow downward phase progression in both temperature and winds. The horizontal winds are quite strong, with a magnitude of $\sim 100\,\mathrm{m\,s^{-1}}$ toward northeast above 95 km, and a magnitude of $\sim 50\,\mathrm{m\,s^{-1}}$ toward southeast around 85 km, and there is a clam layer around 90 km. Squared buoyancy frequency $N^2$ is calculated from background temperature $T_0$ through:

$$N^2 = \frac{g}{T_0}\left(\frac{\partial T_0}{\partial z} + \frac{g}{c_p}\right), \tag{7}$$

where $g$ is the gravity acceleration constant, $c_p$ is the specific heat at constant pressure.

Larger values of $N^2$ indicate a more statically stable atmosphere, while values of negative $N^2$ imply an unstable atmosphere. The squared buoyancy frequencies $N^2$ shown in Figure 5(d) reveal that the background atmosphere was layered with





a convectively stable layer between 90 and 97 km, with larger $N^2$ values about 6–8$\times 10^{-4}$ s$^{-2}$. This stable layer gradually moved downward as modulated by tides. There were multiple relatively unstable layers that existed in between the stable layers around 87 km, 98 km, and 106 km. These layers with positive but smaller $N^2$ can also be unfavorable for wave propagation, as the resulting buoyancy period might be longer than the wave intrinsic period.

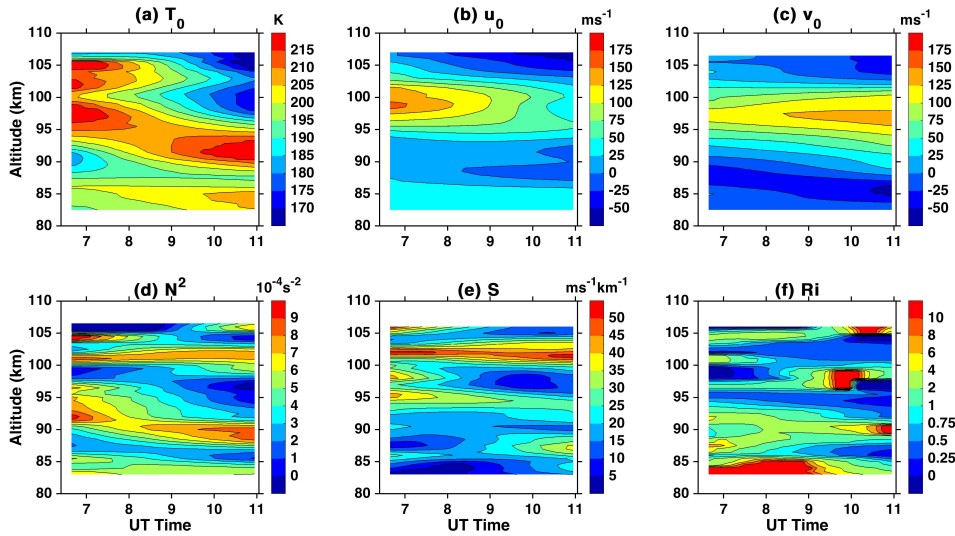

**Figure 5.** Background (a) temperature $T_0$, (b) zonal wind $u_0$ and (c) meridional wind $v_0$. Calculated (d) squared buoyancy frequency $N^2$, (e) vertical shear of horizontal wind $S$ and (f) Richardson number $Ri$. The background here contains all perturbations with periods longer than 6 hour. Note that the positive and negative winds are not symmetric with the colorbar.

Richardson number $Ri$ is commonly used to characterize the dynamical (shear) instability and is calculated through

$$Ri = \frac{N^2}{S^2} = \frac{N^2}{\left(\partial u_0/\partial z\right)^2 + \left(\partial v_0/\partial z\right)^2},$$

(8)

where $S$ is the vertical shear of the horizontal wind. The atmosphere is considered to be dynamically unstable when $0 < Ri < 1/4$. Strong horizontal wind shear and negative vertical temperature gradient make the atmosphere dynamically unstable. As shown in Figure 5(f), the atmosphere is in an overall stable status, with $Ri$ approaching 1/4 in a few thin layers near 87, 95, and 102 km where dynamical instability was likely to occur. Large wind shears are the main factors in these unstable areas.

The layer near 90 km was relatively stable with large $N^2$ and small wind shear; thus a larger $Ri$ where it was favorable for the propagation of the atmospheric wave or allowing wave amplitudes to reach larger values. As shown by the spectral analysis, there are two isolated wave components presented at the same time. The tides mainly dominate the background states shown in Figure 5. However, the long-period (3.2-hr) wave effectively acts as the background for the shorter-period (1.6-hr) wave. The perturbation resulting from the long-period wave might change the temperature gradient, and wind shear leads to a different

stability condition for the shorter-period wave. The background states that include the longer-period (3.2-hr) wave component are attached in the Appendix (Figure A5). Noticeable differences could be identified in the Richardson number where the area





of relatively unstable largely expands, especially above 95 km, which might lead to the dissipation of the shorter-period wave toward higher altitudes. However, the stable layer around 90 km also expands to a slightly wider altitude range, which improves the propagation condition for the shorter-period wave. The corresponding background states are used for the diagnosis of each wave component in later sections.

## 3.2 Filtered Wave Components and Wave Parameters

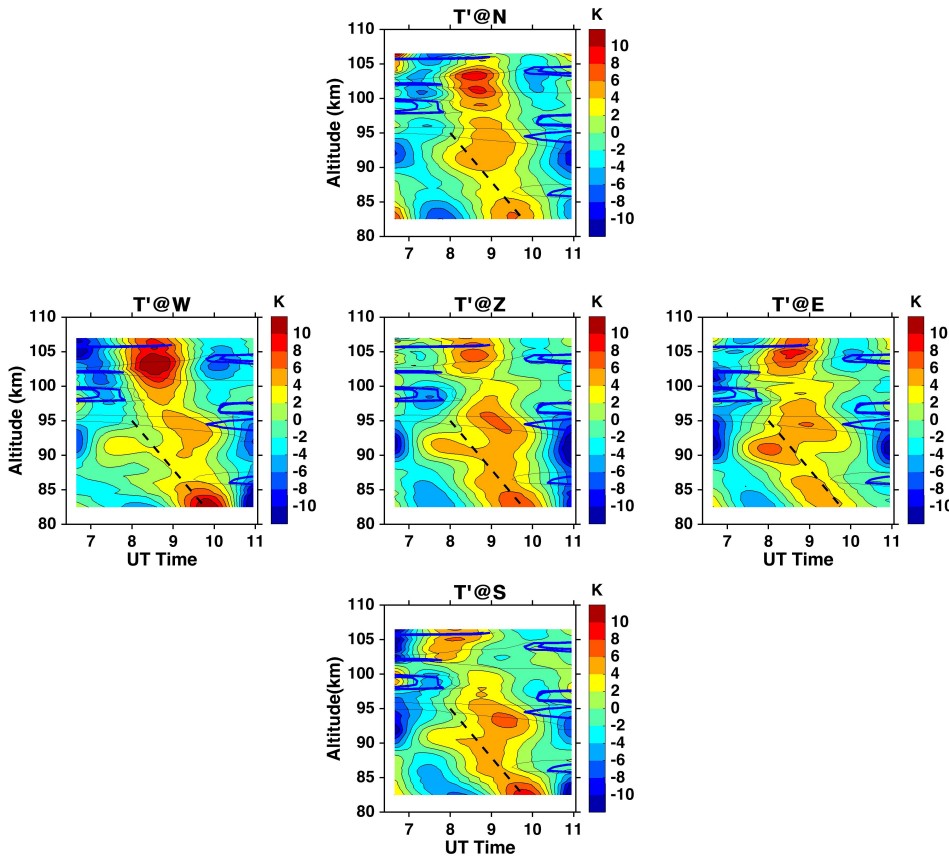

**Figure 6.** Filtered temperature perturbation for the wave #1 (3.35-hr period), in five different directions. Overlapped contours are the Richardson numbers with values 0.25 in blue and 0.5 in black. The dashed black lines mark the downward phase progression and help to distinguish the phase shift in time.

After applying the desired filters on the temperature and wind perturbations, the two dominant wave components are isolated. Using the improved spectral peak determination method, the exact periods of the two dominant components are determined to be 3.35 hr (0.2986 $\text{hr}^{-1}$) and 1.63 hr (0.6148 $\text{hr}^{-1}$). These two wave packets are referred to as wave #1 and wave #2 in the latter analysis. Figure 6 shows the filtered temperature perturbations of wave #1 in all five directions, with Richardson numbers



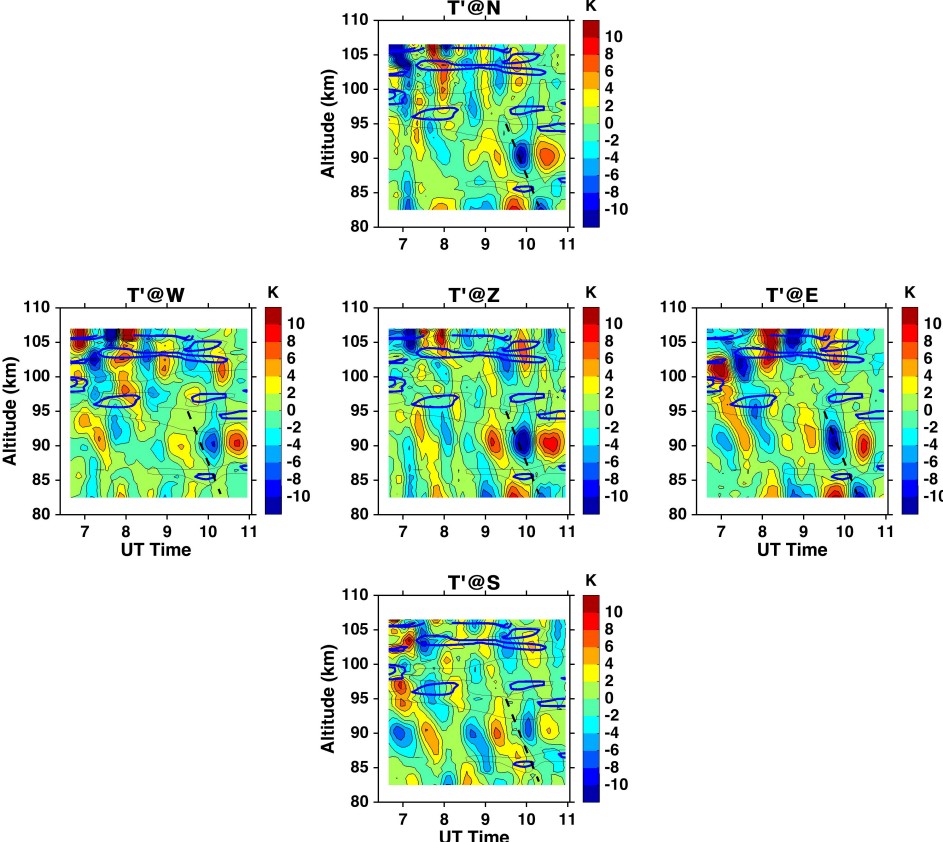

**Figure 7.** Filtered temperature perturbation for wave #2 (1.63-hr period), in five different directions. Overlapped contours are the Richardson numbers with values 0.25 in blue and 0.5 in black. The dashed black lines mark the downward phase progression and distinguish the phase shift in time.

overlapped by contour lines for some values. The atmosphere is generally in a stable condition favorable for upward wave propagation, with only a few thin layers with a Richardson number less than 0.25, and the layers only persist for a short time. However, the wave perturbations have larger amplitudes (about ±5 K) around these layers with a larger Richard number, which makes the waves show up like nodal structures. The phase shift is visible in the perturbations among different directions, the

wave packet roughly moves from the southeast toward the northwest. Figure 7 shows the filtered perturbations for wave #2. Wave pattern and downward phase progression are visible in all directions. The perturbations reach a maximum of ±10 K at about 90 km altitude, confined within both layers above and below with smaller magnitudes of Richardson numbers. Another maximum can be seen at below 85 km. Even though there are strong wave patterns at higher altitudes, the wave pattern is less consistent among different directions and shows up with various periods. At this unstable layer, nonlinear wave-mean flow

interaction might exist, resulting in the dispersion of the wave packet. This is also shown by the spectra in Figure 4, where the broader spectra at the higher altitudes indicate the dispersion of wave packets.



Filtered wave components in wind perturbations are shown in the Appendix (Figures A3 and A4). The wave signatures of both components are evident in the horizontal winds with the visible downward phase progression, and the node structure is clear in the vertical direction with at least two maxima at different altitudes. The wave patterns are slightly different in zonal,

meridional and vertical winds as the relations are determined by the polarization formulas. Even though the wave pattern is clear, the phase shifts among measurements of different directions are imperceptible in horizontal wind perturbations for both wave packets. The measurement uncertainties of 5–10 $\mathrm{m\,s^{-1}}$ are too large compared to the wave amplitudes of 10–20 $\mathrm{m\,s^{-1}}$, making the slight phase shift hard to be distinguished. The assumption of vertical wind homogeneity without considering the phase shift also aggravates the uncertainties of horizontal winds. In later analysis, only the temperature measurements are used

for the cross-spectral method to estimate wave parameters.

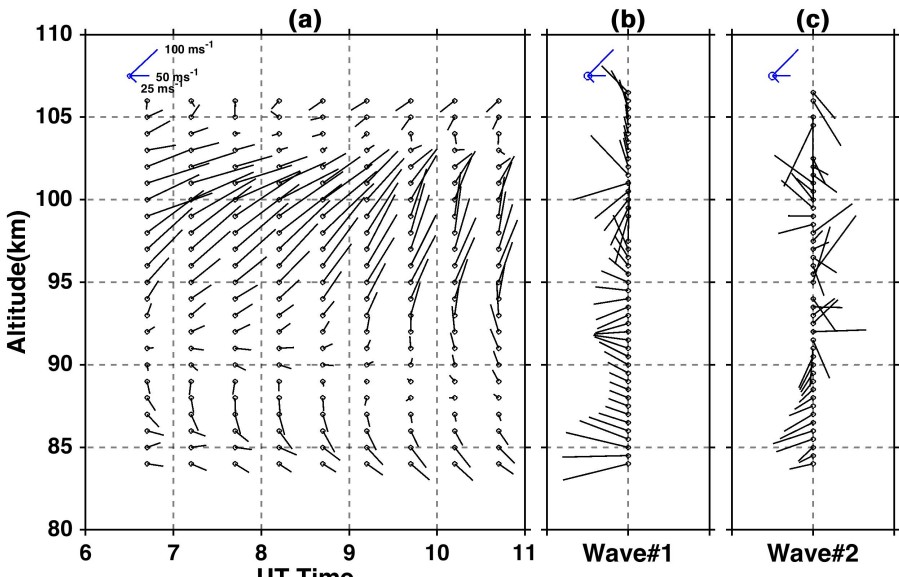

**Figure 8.** (a) Background winds, (b)horizontal phase speed of wave #1, and (c) horizontal phase speed of wave #2, the staff pointing to the directions that wind blows to or wave propagates toward, and the length representing the wind speed or phase speed magnitudes. The ratio of the axis scale (5-km/1-hr) matches the zonal and meridional components of wind and wave speeds, to let the staff point to the right direction on the plane. The legends in the upper-left corner of each panel denote the speeds of 100 $\mathrm{m\,s^{-1}}$, 50 $\mathrm{m\,s^{-1}}$, and 25 $\mathrm{m\,s^{-1}}$ at NE, E and SN directions, respectively.

For a reliable estimate of the phase shift, we choose the temperature measurements at four off-zenith directions to derive horizontal wave information. Using the proposed cross-spectral method, the phase shifts in zonal and meridional directions were firstly derived, and horizontal wavelength, propagation azimuth, and observed phase speed were then calculated. In Figure 8, the background wind and wave phase speed at different altitudes are shown in the 'wind barb' manner, with the

length representing speed magnitudes and the staff pointing to the direction the wind blows to and the wave propagates toward. In Figure 8(b) and 8(c), no wind barb is shown at some altitudes. This is because the derived wave phase speed is too large





$(c - u_0 > 0.5c_s)$ (Zhou and Morton, 2007), invalidating the results. Here, the speed of sound $c_s$ in the atmosphere is estimated as $c_s = \sqrt{\gamma R T_0}$, where $\gamma$ is the ratio of specific heat, $R$ is the ideal gas constant, and $T_0$ is the background temperature. The typical speed of sound $c_s$ is calculated to be around $280 \mathrm{~m\,s^{-1}}$ in the lidar observation altitude range. During the lidar

observation period and altitude range, the background wind is mostly toward the east, with strong northeastward winds above 95 km and moderate southeastward winds around 85 km. The determined propagation azimuth and phase speed are relatively consistent between 85 km and 100 km for wave #1, and below 90 km for wave #2. And both wave packets are estimated to propagate toward the west, with wave #1 propagating mostly toward $W$-$NW$, and wave #2 propagating toward $W$-$SW$. In ideal conditions, these ground-observed wave parameters are invariant if the wave pattern sustains and does not dissipate. The

parameters and their uncertainties, represented by the mean and standard derivation of propagation azimuth, phase speed, and horizontal wavelength within the most reliable altitude range, are listed in Table 1. Both frequency/period and phase speed are observed in a ground-based frame. The horizontal wavelength wave #1 and wave #2 are estimated to be around 975 km and 438 km, both with a $\sim$20 % uncertainty. The propagation azimuth angles are estimated to be 299° and 233° for two waves, both with a 15°–20° uncertainties. These wavelengths and azimuths correspond to phase shifts of -32° and 18° between measurements

of $E$-$W$ and $N$-$S$ for wave #1, and phase shifts of -65° and -49° for wave #2. The observed phase speeds of two wave packets are quite similar to be around $80 \mathrm{~m\,s^{-1}}$. Both waves propagate through the background atmosphere with varying stability and potentially undergo some dissipation and dispersion, especially at higher altitudes. Therefore, the monochromatic wave assumption is no longer satisfied there, and wave speed and azimuth determined from the phase shift show large fluctuations at these altitudes. The wave pattern shows downward phase progression in the vertical directions, and not a single complete

wave cycle is identified due to larger vertical wavelengths. Therefore, the vertical wavelengths were roughly estimated from the phase slope to be around 22 km and 24 km for the two wave packets, which are larger than the valid altitude range of lidar measurements.

**Table 1.** Gravity wave parameters retrieved from lidar measurements in the ground-based observing frame.

| Wave | Frequency(hr$^{-1}$) | Period(hr) | Azimuth(°)[a] | H. Wavelength(km) | Phase Speed(m s$^{-1}$) | V. Wavelength(km)[b] |
|------|------|------|------|------|------|------|
| #1 | 0.2986±0.055 | 3.35±0.6 | 299±17 | 975±260 | 85±22 | $\sim$22 |
| #2 | 0.6148±0.094 | 1.63±0.3 | 233±15 | 438±135 | 76±23 | $\sim$24 |

[a] The azimuth angle is measured clockwise from the North.

[b] The vertical wavelength was estimated from the downward phase progression.

## 3.3 Wave Diagnosis: Dispersion and Polarization Relations

Gravity wave dispersion relation links the vertical wavenumber to the horizontal wave parameters and background states. It

is often used to diagnose gravity wave propagation, reflection, and ducting. For the acoustic-gravity waves in a compressible atmosphere, the equations (9) and (10) in Zhou and Morton (2007) are full descriptions of the dispersion relation. For waves with a small intrinsic horizontal phase speed ($|c - \overline{u}| < 0.5c_s$), which is valid for most observed gravity waves, the dispersion





relation can be described as:

$$
\begin{aligned}
m^2 &= \frac{N^2}{(c-\overline{u})^2} - k_H^2 - \frac{1}{4H_s^2} + \frac{1}{c-\overline{u}}\frac{d^2\overline{u}}{dz^2} + \frac{2-\gamma}{\gamma}\frac{1}{H_s(c-\overline{u})}\frac{d\overline{u}}{dz} - \frac{3}{c_s^2}\left(\frac{d\overline{u}}{dz}\right)^2 \\
&+ \frac{g}{H_s(c-\overline{u})^2}\frac{dH_s}{dz} + \frac{1}{2H_s}\frac{d^2H_s}{dz^2} - \frac{3}{4}\left(\frac{1}{H_s}\frac{dH_s}{dz}\right)^2 - \frac{1}{H_s(c-\overline{u})}\frac{d\overline{u}}{dz}\frac{dH_s}{dz},
\end{aligned}
\tag{9}
$$

where $H_s = RT_0/g$ is the density scale height and $\gamma$ is the ratio of specific heat, and $c$, $\overline{u}$ and $c_s$ are observed horizontal phase

speed, background wind speed in the direction of wave propagation and speed of sound, respectively. Horizontal wavenumber $k_H$ is related to $k$ and $l$ through $k_H^2 = k^2 + l^2$. The term $c - \overline{u}$ is the intrinsic horizontal phase speed, usually denoted as $\hat{c}$. The corresponding intrinsic wave frequency is related to observed wave frequency by $\hat{\omega} = \omega - k_H \cdot \overline{u}$. When the atmosphere is treated as incompressible and background temperature varies slowly within the vertical wavelength of the wave, we have $c_s \to \infty$ and $dH_s/dz \to 0$. The dispersion relation (9) is reduced to the following form that is derived based on Taylor-Goldstein equation

(Nappo, 2012, equation 2.34):

$$
m^2 = \frac{N^2}{(c-\overline{u})^2} - k_H^2 - \frac{1}{4H_s^2} + \frac{1}{(c-\overline{u})}\frac{d^2\overline{u}}{dz^2} + \frac{2-\gamma}{\gamma}\frac{1}{H_s(c-\overline{u})}\frac{d\overline{u}}{dz}.
\tag{10}
$$

The coefficient of the last term in equation (10) is different from the original one due to a correction for a compressible atmosphere based on discussions in Zhou and Morton (2007). If the wind shear terms are further neglected, the dispersion relation (10) is simplified to equation (24) in Fritts and Alexander (2003) but without the Coriolis term and is also same as the dispersion relation derived by Hines (1960)

$$
m^2 = \frac{N^2}{(c-\overline{u})^2} - k_H^2 - \frac{1}{4H_s^2}.
\tag{11}
$$

Through the dispersion equations, the vertical wavenumber $m$ is related to the wave characteristics, including the horizontal wavenumbers $k_H$ and phase speed $c$, and the background states including the projected wind on wave propagation direction $\overline{u}$ and wind shear, and background temperature $T_0$ and its gradient as reflected by scale height $H_s$ and buoyancy frequency $N$. In the regions of the atmosphere where $m^2 > 0$, gravity waves are able to propagate freely and are characterized by corresponding $m$, $k$, $l$, and $c$. Regions of $m^2 < 0$ indicate evanescence for gravity waves, where wave amplitudes decay exponentially. When

a propagating wave encounters a region where $m^2 < 0$, partial or total reflection can occur depending on the depth of the evanescent region. Gravity waves whose propagation is restricted in a region with reflective regions of evanescence, above and below, are said to be ducted. At altitudes where $c = \overline{u}$, the vertical wavenumber approaches infinity, and the waveform is overturned, so the wave breaking or dissipation occurs. The phenomenon is called wave critical-layer filtering, which could be partially or entirely. In this case study, the altitude range is limited to be within 85–105 km, during which the background

temperature gradient is about -5 K km$^{-1}$ and the wind shear is strong as 30 m s$^{-1}$ km$^{-1}$. The contribution of terms related to the temperature gradient and wind shear is not insignificant. Since all wave parameters and background states are explicitly determined, we evaluated all three forms of dispersion relations in the diagnostics of the propagation of the identified wave packets.

Figure 9(a) and 9(d) show the calculated $m^2$ for the two retrieved wave packets, based on the full dispersion relation of

equation (9). In general, the $m^2$ shows up in layered structures for both waves, potentially creating ducts for the gravity waves.





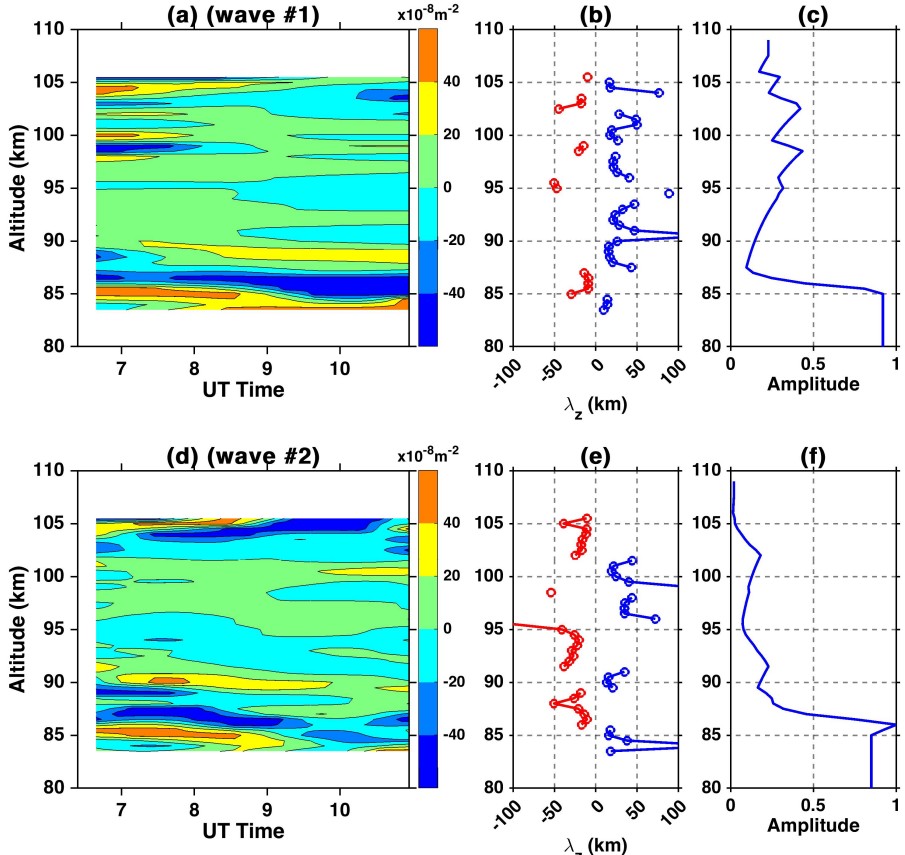

**Figure 9.** Calculated $m^2$ for wave (a)#1 (3.35-hr) and (d)#2 (1.63-hr) using equation (9), and the vertical wavelength derived from the mean $m^2$ for wave (b)#1 (3.35-hr) and (e)#2 (1.63-hr). The negative wavelengths correspond to the evanescent regions. The relative wave amplitude estimated from the decay and growth rate for wave (b)#1 (3.35-hr) and (e)#2 (1.63-hr).

For the 3.35-hr wave (#1), there are major layers of negative $m^2$ around 85–86 km, 95 km, 98 km, and 103 km altitudes. There are also several thin layers of negative $m^2$ at other altitudes lasting shorter times. When upward propagating waves encounter the layers, their amplitudes will attenuate on top of the growth due to decreased atmospheric density. For the evanescent layer around 85–86 km, the waves are supposed to dampen at a fast rate (larger value of $im$). However, this layer is thin with a

maximum thickness of 2 km, plenty of wave energy could penetrate the layer to higher altitudes. This also applies to other thin evanescent layers above where the attenuation is even less. The upward propagating waves were partially reflected and refracted at each evanescent layer, which could change the wavefront orientation. This partial refraction/reflection was also presented by the previous numerical simulations in similar scenarios (Heale et al., 2014; Cao et al., 2016). For the temperature perturbation in Figure 6, the wave pattern shows up with a nodal structure with clear amplitude maximums around 85 km,

92 km, and 102 km, and discontinuities in the phase progression in the vertical direction. These features directly result from





the partial reflection and refraction at the multiple evanescent layers where the perturbations observed by the lidar are the superposition of incident and reflected waves between the evanescent layers. For the 1.63-hr wave (#2), the overall layered structure in $m^2$ remains similar to the 3.35-hr wave. However, the 3.35-hr wave effectively changes wind and temperature and creates differences in $m^2$, and leads to different propagation conditions for the 1.63-hr wave. The evanescent layer around

86 km reduced the attenuation rate from 09:00 UT onwards, which largely increased the portion of the transmission of wave energy. And 3.35-hr wave increases the evanescent layer thickness to 5 km around 95 km, which could limit the transmissible wave energy to further altitudes. As shown in Figure 7, there is a visible maximum below 85 km, and another maximum was around 90 km whose wave amplitudes largely increased after 09:00 UT. The amplitudes decreased dramatically right above 95 km, however, some wave peaks can be seen above 100 km. The average $m^2$ profiles of the whole period are calculated

for both wave packets, the corresponding vertical wavelengths are estimated and shown in Figure 9(b) and 9(d). The positive wavelengths for the freely propagation wave are about 25 km, closely matching the ones estimated from the downward phase progression slope. The negative wavelength corresponds to the evanescent region and is equivalent to the scale height of the wave amplitude decay. The overall decay/growth rate of wave amplitude is described by $e^{(1/2H_s - im)z}$, where delay occurs when $m$ is an imagery number in the evanescent region. In Figure 9(c) and 9(f), the relative wave amplitudes are estimated,

assuming a unit amplitude at the lowest altitude. The predicted wave amplitude shows fluctuation and several maxima at different altitudes, which are the combined efforts of evanescent decay and conserved growth. Note that the evanescent layers at around 86 km are supposed to attenuate the wave amplitude a lot (by 70–80 %), but layers with the time-varying $m^2$ could reduce the dampening efforts and spare more energy penetrating to higher altitudes. Bossert et al. (2014) presents case studies using lidar observations and simulations to show high-frequency (period shorter than 15 min) gravity waves propagating to

higher altitudes through alternating regions of evanescence and freely propagation over a few kilometers. In this study, the two medium-frequency wave packets observed by the lidar propagate through multiple thin and time-varying evanescent layers with a good portion of wave energy penetrated to higher altitudes, and partial reflection and refraction occur with the observed amplitude maxima found between these evanescent layers.

     The $m^2$ estimated by equations (10) and (11) are shown in the Appendix (Figures A6 and A7). The results of equation

(10) show overall similarity with the ones of equation (9), but some differences exist. The evanescent layers estimated by equation (10) are slightly thinner, which shows the contribution of neglected temperature gradients. The $m^2$ calculated by equation (11) fails to capture most of the layered structures and underestimates all evanescent layer thickness as wind shear is a significant contributing factor for both waves. As discussed above, inconsistency exists among different dispersion relations, and some simplifications fail to capture the authentic characters. In the application of dispersion relations, the full background

temperature and wind measurements might not be available in all cases or are sometimes limited only to a few altitudes such as airglow layers. Nevertheless, simplified formulas (equations (10) and (11)) can be best utilized to diagnose wave propagation; however, the results should be interpreted with caution.

     Another important relation derived from linearized wave equations is the polarization relation that describes the relative phase differences and amplitude ratios of various wave quantities. If gravity waves do not undergo dissipation, the complex

wave amplitudes of the relative temperature $\tilde{T}\left(T'/\overline{T}\right)$, zonal wind $\tilde{u}$, meridional wind $\tilde{v}$ and vertical wind $\tilde{w}$ should satisfy the





following polarization relations (Fritts and Alexander, 2003; Vadas, 2013; Lu et al., 2015):

$$\frac{\tilde{T}}{\tilde{w}} = \frac{N^2\left(im + \frac{1}{2H_s}\right) - \frac{\hat{\omega}^2}{\gamma H_s}(\gamma - 1)}{g\hat{\omega}\left(-m - \frac{i}{2H_s} + \frac{i}{\gamma H_s}\right)}$$

$$\frac{\tilde{T}}{\tilde{u}} = \frac{N^2\left(im + \frac{1}{2H_s}\right) - \frac{\hat{\omega}^2}{\gamma H_s}(\gamma - 1)}{g} \frac{\left(\hat{\omega}^2 - f^2\right)(k\hat{\omega} - ifl)}{\left(N^2 - \hat{\omega}^2\right)(k^2\hat{\omega}^2 + f^2l^2)} \qquad (12)$$

$$\frac{\tilde{u}}{\tilde{v}} = \frac{i\hat{\omega}k - fl}{i\hat{\omega}l + fk}.$$

The complex amplitudes of $\tilde{T}/\tilde{w}$, $\tilde{T}/\tilde{u}$ and $\tilde{u}/\tilde{v}$ describe the amplitude and phase relations among different wave quantities. On the one hand, the missing quantities of observed gravity waves can be estimated through these relations assuming non-dissipation. On the other hand, the discrepancies between observed and theoretical values can be used to indicate wave dissipation. It is also possible to estimate higher-order statistical quantities, such as gravity wave momentum flux ($\overline{u'w'}$) and heat flux ($\overline{w'T'}$) from these relationships with limited observations (Liu, 2009; Guo et al., 2017).

**Table 2.** Amplitude ($A$) ratio and phase ($\phi$) difference between quantities of $\tilde{T}$ and $\tilde{w}$, $\tilde{T}$ and $\tilde{u}$, $\tilde{u}$ and $\tilde{v}$.

| Quantities | $\frac{A(\tilde{T})}{A(\tilde{u})}$[a] | $\phi(\tilde{T}) - \phi(\tilde{u})$ | $\frac{A(\tilde{T})}{A(\tilde{w})}$ | $\phi(\tilde{T}) - \phi(\tilde{w})$ | $\frac{A(\tilde{u})}{A(\tilde{v})}$ | $\phi(\tilde{u}) - \phi(\tilde{v})$ |
| --- | --- | --- | --- | --- | --- | --- |
| Units | %m$^{-1}$s | Deg. | %m$^{-1}$s | Deg. | NaN | Deg. |
| Wave #1 Propagating[b] | 0.32±0.13 | -109±12.81 | 9.62±5.1 | -101.6±4.8 | 1.77±0.1 | 166.57±6.0 |
| Wave #1 Evenescent[b] | 0.20±0.13 | -9.67±46.80 | 5.32±3.9 | -66.0±63.34 | 1.77±0.1 | 166.57±6.0 |
| Data[c] | 0.21±0.16 | | 3.92±2.53 | | 1.03±0.86 | |
| Wave #2 Propagating[b] | 0.48±0.21 | -116.11±14.5 | 3.28±2.36 | -102.42±5.3 | 1.33±0.1 | 4.47±2.19 |
| Wave #2 Evenescent[b] | 0.26±0.13 | 5.42±36.22 | 2.47±1.30 | -82.80±36.0 | 1.33±0.1 | 4.47±2.19 |
| Data[c] | 0.25±0.22 | | 1.52±1.34 | | 1.02±0.52 | |

[a] $\tilde{T}$ here is relative temperature perturbation $T'/\overline{T}$ expressed in percentage.

[b] The mean and standard derivation of the quantities are calculated within corresponding altitudes range.

[c] The altitude range of 85–100 km was used to calculate the mean and standard deviation.

The theoretical values of $\tilde{T}/\tilde{w}$, $\tilde{T}/\tilde{u}$ and $\tilde{u}/\tilde{v}$ can be calculated from equation (12) with the retrieved wave and background parameters. They are complex numbers, with their absolute magnitudes representing the wave amplitudes ($A$) ratio and phases representing the phase ($\varphi$) difference between any two quantities. There are two sets of amplitude ratios and phase differences are calculated, one for positive $m^2$ and the other for negative $m^2$, which correspond to the wave free-propagating and evanescent regions. Table 2 lists the theoretical results for both wave packets. In the region of free-propagating, the values are relatively constant; the variations are mainly because of the change of intrinsic frequency along altitudes. However, large variations exist for the phase differences in the evanescent region. In this case, the wave packets propagate through multiple evanescent layers where partial reflection occurs. The lidar observed perturbations are the superposition of incident and reflected waves and the propagating waves could undergo dissipation. It is difficult to accurately estimate the amplitude ratios





and phase differences from the observed perturbations. Another difficulty lies in that the horizontal winds are not retrieved at the zenith direction and have to be indirectly estimated from horizontal winds at off-zenith directions by correcting the phase shift. Therefore, we only estimate the averaged amplitude ratios from the observations. In Table 2, the observational results are estimated with larger uncertainties, and all show discrepancies with the predicted ones. However, actual values of $A(\tilde{T})/A(\tilde{u})$

are closer to the predicted evanescent ones. Besides the uncertainties in the measurements, especially winds, wave dissipation could also contribute to it. For the ratio $A(\tilde{T})/A(\tilde{w})$, it is much smaller than the predicted values, which means the observed perturbations in the vertical wind might contain large errors. The presented results reveal the complexity of gravity waves propagating in the atmosphere. The polarization relation is good for diagnosing the free propagating waves without being reflected and refracted, and dissipation is not severe. It might not be proper when complicated wave-mean flow interaction occurs, such

as in this wave case.

Using the proposed cross-spectral method, we identified two gravity wave packets and retrieved all the wave parameters, and determined the background states. The fully retrieved information was used to validate the linear gravity wave theories with the least assumptions. Consistent results are obtained from the diagnostic analyses, which mostly explain the wave observations. However, raw lidar measurement uncertainties exert difficulties in some wave parameter estimations and diagnostic analyses.

In the next section, we implemented a sensitivity study to evaluate the general usage of this method in detecting gravity waves.

## 4  Sensitivity Study

It is well-known that most observation techniques are restricted by the 'observation filter' effect in resolving atmospheric waves. These techniques are sensitive to certain parts of the spatial and temporal spectra of the waves. The effect also applies to the lidar and the wave extraction method presented in this study. To find out a spectral range of gravity waves that is

more favorable to be identified by this method, we did a sensitivity study using a forward simulation. The accuracy of the cross-spectral methods in recovering the wave parameters of amplitude, period, and phase shift is quantified.

The off-zenith angle of laser beams coupled with steerable telescopes could be adjusted. However, this configuration is no longer available, and newer lidar systems are equipped with multiple fixed telescopes pointing to certain directions; thus the off-zenith direction and angle are fixed. The photon integration time at each direction and the laser beam rotating sequence

could be changed based on applications, such as only one direction in zonal and meridional directions. There are also some lidar systems with one master laser beam being split and shooting to multiple directions simultaneously. The typical photon integration time at each direction is around one to several minutes. A configuration similar to the lidar deployed in Maui is utilized in the simulation where a 30° off-zenith angle corresponds to a separation of ~100 km at 90 km altitude between two off-zenith directions ($W$ and $E$, or $S$ and $N$). For simplicity, the temporal resolution was selected to be a uniform 6-min

in all directions. Because the retrieval of wave parameters in zonal and meridional directions ($k$ and $l$) can be independent, the simulations only consider measurements from two laser beams aligned in any direction. The perturbations of a traveling





gravity wave observed by two laser beams are described by

$$y_1 = A \cdot \sin(\omega \cdot t + \phi_1) + \delta_1$$
$$y_2 = A \cdot \sin(\omega \cdot t + \phi_2) + \delta_2$$
(13)

of which the definitions of the terms are the same as equation (4). Extra terms $\delta_1$ and $\delta_2$, two independent (associated with different 'seeds' of the pseudorandom number generator) sets of Gaussian-distributed random numbers, are introduced to

mimic the uncertainties of measurements. The cross-spectral method is applied on the data $(y_1, y_2)$ to retrieve the wave amplitude $A$, period $\tau = 2\pi/\omega$, and phase shift $\Delta\phi = \phi_1 - \phi_2$. Then, horizontal wavelength/wavenumber and phase speed are further estimated. To qualify the accuracy, the percentage errors of wave amplitude $A$, period $\tau$, and phase difference $\Delta\phi$ between retrieved results and preset values are calculated.

In the simulation, the wave amplitude $A$ is selected as two (with an arbitrary unit) and the random noise $\delta$ has a standard

deviation of 0.5. Therefore, the following simulations and discussions only apply to this ratio of wave amplitude to noise ($A/\delta$). Five hours of data (50 data points) were used in the simulation. In order to recover the spectral peaks falling between two integer spectral points, the aforementioned fitting method was used to best retrieve the actual peaks. The horizontal wavelength is selected to vary from 300 km to 3000 km (3 to 30 times the separation), and the period range is from 1 hr to 5 hr. In Figure 10, the percentage errors of wave amplitude, period, and phase shift are shown. The retrieved wave amplitudes have an error of 10 %

from the true value for all periods and wavelengths. A slight negative bias of 5 % and a wave pattern are identified in amplitude errors. The retrieved period is very close to the true value, with a 5 % error for periods shorter than 3-hr. And the errors in periods also show some wave patterns and errors are much larger for waves with longer periods and shorter wavelengths. The retrieved phase shift shows an error of within 10 % for all periods and wavelengths shorter than 2000 km. The cross-spectral method is unable to retrieve the small phase shifts (12°–18°) resulting from waves with very long horizontal wavelengths

(2000–3000 km), showing errors up to 60 %. In this simple simulation, there is only one frequency being simulated. If spectral leakage is not an issue when the actual peak is close to the integer spectral points, the wave parameters could be determined accurately. However, with limited samples, the resolution in the spectral domain is coarse, and there is a good chance of spectral leakage that the actual peaks fall between two integer spectral points, especially around lower frequencies. The negative bias of the retrieved amplitude and the wave pattern in the errors of three parameters are related to spectral leaks. As lidars usually

provide measurements over an altitude range between 80 km and 110 km, the wave propagating in the vertical direction can be captured. The retrieved period, wavelength, and phase speed are invariant if the waves do not undergo severe dissipation. This could provide another level of verification of the retrieved wave parameters at different altitudes.

As shown in the simulation results, the proposed method relies on cross-spectral analysis and inherits the typical drawback of any spectral method. Even though a parabolic fitting is used to improve the determination of the spectral peak, it does not

recover the peaks accurately when the leakage is severe. The cross-spectral method can identify the wave amplitude and period with an estimated accuracy of better than 10 % and phase shifts with a 20 % uncertainty. The retrieved wave amplitudes are slightly smaller (5 %) than the true values. The errors in retrieved periods are slightly larger for waves with longer periods (close to data duration). The errors of both periods and phase shifts will propagate to the further derived wavelength and





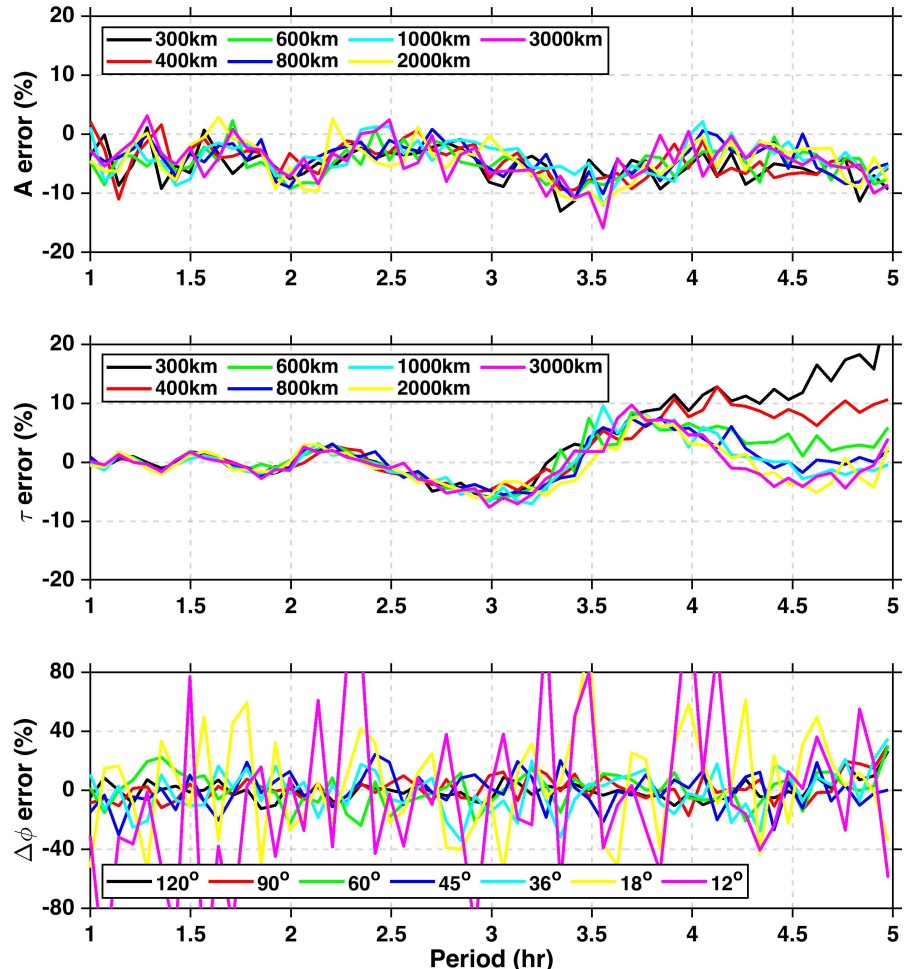

**Figure 10.** Percentage errors of (top) wave amplitude $A$, (middle) period $\tau$, (bottom) phase shift $\Delta\phi$ between the retrieved results and preset values, for gravity wave packets with different periods (1–5 hr) and wavelengths (300–3000 km) but with fixed amplitude and added random noise. The phase shifts $\Delta\phi$ result from a 100 km separation between two laser beams.

phase speed. When multiple wave components are mixed together, the spectral leakage will be more complicated and some
filtering is necessary before the cross-spectra analysis. With measurements from only two directions, the retrieved wavelength
is an apparent one along that direction. The actual wavelength of a 2-D wave propagating in any direction can be retrieved by
combining wavelengths projected to zonal and meridional directions.

Here, we provide an introductory assessment of the proposed wave extraction method. The data duration is a critical factor;
more data points could help alleviate the spectral leakage and retrieve accurate wave parameters. However, the duration is
limited by the operation of such a lidar that mostly works at night time unless extra filters are used to facilitate the daytime
operation (Chen et al., 1996). And there is a trade-off between longer datasets for higher spectral resolution and the duration



of wave presence with an invariant period and amplitude. The most favorable wave period of this method is limited to be less than data duration to minimize the spectral leakage and much longer than the temporal resolution to achieve enough data points to resolve the variation in one wave cycle. The phase shifts need to be large enough to be distinguished by the cross-spectral method, which makes the method unsuitable for waves with very long wavelengths. However, this is subject to this wave amplitude to noise ratio ($A/\delta$). If the observed perturbations are associated with larger ratios, the method should be able to identify smaller phase shifts and determine longer wavelengths. It is hard to give an applicable wavelength range for this method, as it is highly subject to the actual data quality, i.e., the signal-to-noise ratio and off-zenith angle. Such a sensitivity analysis can be implemented for specific lidar system configurations. As a practical remark, the proposed method is adept at detecting gravity waves of medium-scale and medium-frequency for a lidar operated at similar settings and with comparable measurement uncertainties.

## 5   Discussions and Summary

With measurements from a single ground-based instrument, a narrow-band sodium lidar operated in multiple-direction mode, two gravity wave packets were detected on the night of 14 January 2002 at Maui, Hawaii, and were resolved in 3-D space by a cross-spectral method. The detected phase differences among measurements in different directions enable the retrieval of the horizontal information of the wave. Using this method, the horizontal wavelength and phase speed are estimated with $\sim$20 % uncertainties. One wave with a horizontal wavelength of 975 km and a period of 3.35 hr propagated toward 299° azimuth at a phase speed of $85\,\mathrm{m\,s^{-1}}$, the other one with a horizontal wavelength of 438 km and a period of 1.63 hr propagated toward 233° azimuth at a phase speed of $76\,\mathrm{m\,s^{-1}}$. Both waves propagated toward the nearly opposite direction of the background winds and larger wave amplitudes were found in the relatively stable regions, as indicated by the squared buoyancy frequency $N^2$ and Richardson number $Ri$. With full sets of wave parameters and background states determined, multiple versions of dispersion relation, some with simplifications, are examined in this study. Both wave packets are found to propagate through multiple thin evanescent layers where partial reflection and refraction occur around those evanescent layers. However, a good portion of wave energy penetrates to higher altitudes where waves undergo further dissipation and dispersion. The longer-period (3.35-hr) wave effectively changes the background and leads to a different propagation condition for the shorter-period (1.63-hr) wave. The comparisons among different versions of dispersion relations show that the effects of background temperature gradient and wind shear are important in the linearized wave theory, and diagnostic analysis based on simplified dispersion relations should be interpreted with caution. Polarization relations are also examined among terms $\tilde{T}$, $\tilde{u}$, $\tilde{v}$ and $\tilde{w}$. However, the complexity of the wave propagation conditions and uncertainties of the measurements, especially in the winds, make it difficult to retrieve accurate results from the data.

Continuous lidar measurement profiles with proper sampling rate and duration can capture a wide variety of periods of waves. However, those profiles are normally only used to resolve the vertical variations and horizontal information is often complemented by other observations. In this study, we propose a novel method that uses a single-site lidar configured in multiple-direction observing mode to resolve gravity waves fully in 3-D space. The sensitivity study reveals the capability

of this method in detecting medium-scale and medium-frequency gravity waves. This partially makes up the spectral gaps in the 'observation filter' mentioned above, with horizontal and vertical wave information both retrieved directly. The proposed method could provide extra opportunities for the gravity wave studies based on lidar systems with similar configurations that were deployed at other sites, either in the past or still in operation (Hu et al., 2002; Hildebrand et al., 2012; Cai et al., 2014; Ban et al., 2015; Liu et al., 2016; Li et al., 2020). Unlike the lidar presented in this study, newer lidar systems are equipped

with 2–4 telescopes that are pointed in several directions depending on research requirements, and the off-zenith angles are often fixed. If their sampling interval and rotating sequence are properly configured, it is feasible to use this method to detect more medium-scale and medium-frequency gravity wave events. The determination of 3-D wave parameters combined with background atmosphere states would also enable backward ray tracing to identify the wave source location (Vadas et al., 2009; Krisch et al., 2017; Krasauskas et al., 2023). To further examine the wave propagation, reflection, and dissipation, a numerical

simulation that takes account of the complete wave parameters and background states would provide important insights into the interpretation of observed results and unobserved beyond field-of-view.

*Data availability.*  The narrow-band sodium lidar data, including sodium density, temperature, and winds from the Maui/MALT campaign and other deployments of an upgraded lidar system operated by UIUC and Embry-Riddle Aeronautical University can be found at http://alo.erau.edu/data/nalidar/index.php and the data used for this study can also be downloaded at https://zenodo.org/record/8124900.

*Author contributions.*  AZ processed the raw lidar data and administrated the data curation, BC developed the methodology and analyzed the data. BC prepared the manuscript with the contribution of AZ.

*Competing interests.*  The authors declare that they have no conflict of interest.

*Disclaimer.*  Any opinions, findings, and conclusions or recommendations expressed in this material are those of the author(s) and do not necessarily reflect the views of the National Science Foundation.

*Acknowledgements.*  This work is supported by National Science Foundation (NSF) grants AGS-1110199 and AGS-1115249. The work by Alan Liu is supported by (while serving at) the NSF, USA.



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





## List of Figures









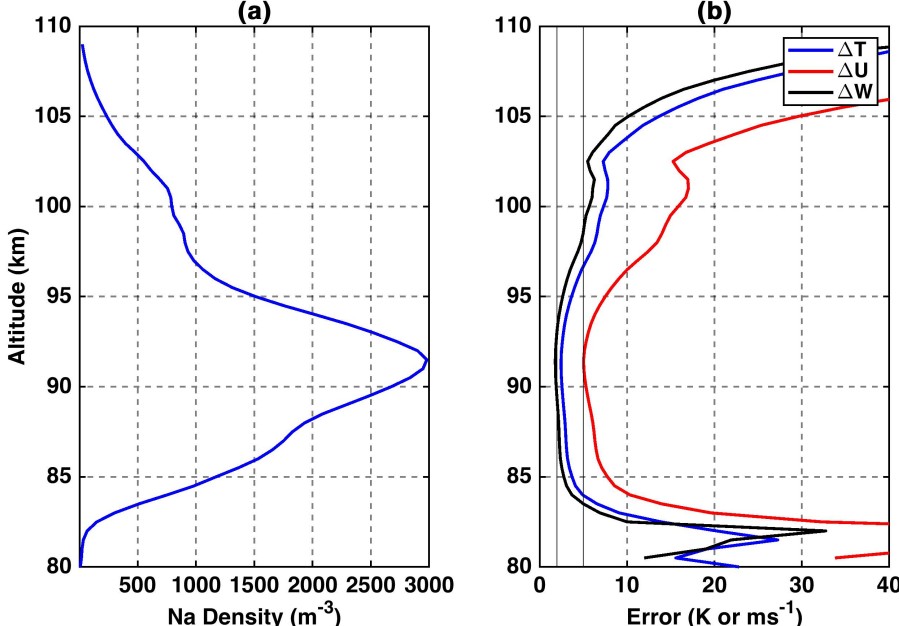

**Figure A1.** (a) Mean sodium density and (b) estimated uncertainties for temperature, horizontal and vertical winds.



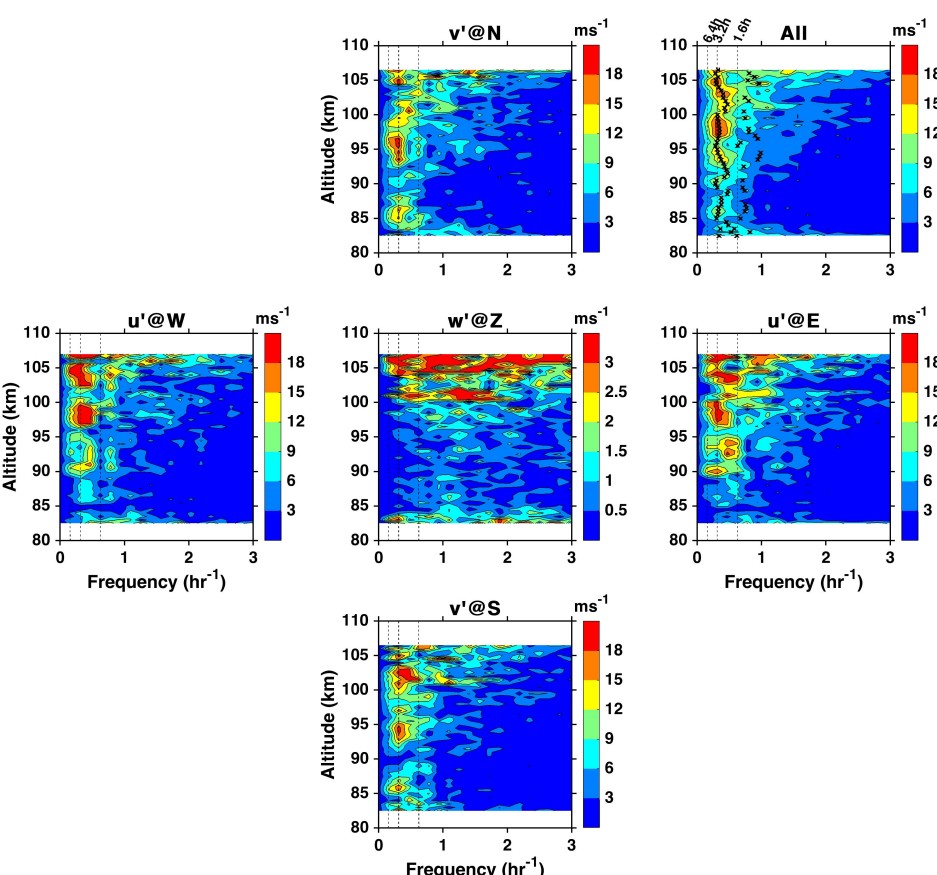

**Figure A2.** Same as Figure 4 but for horizontal and vertical winds.



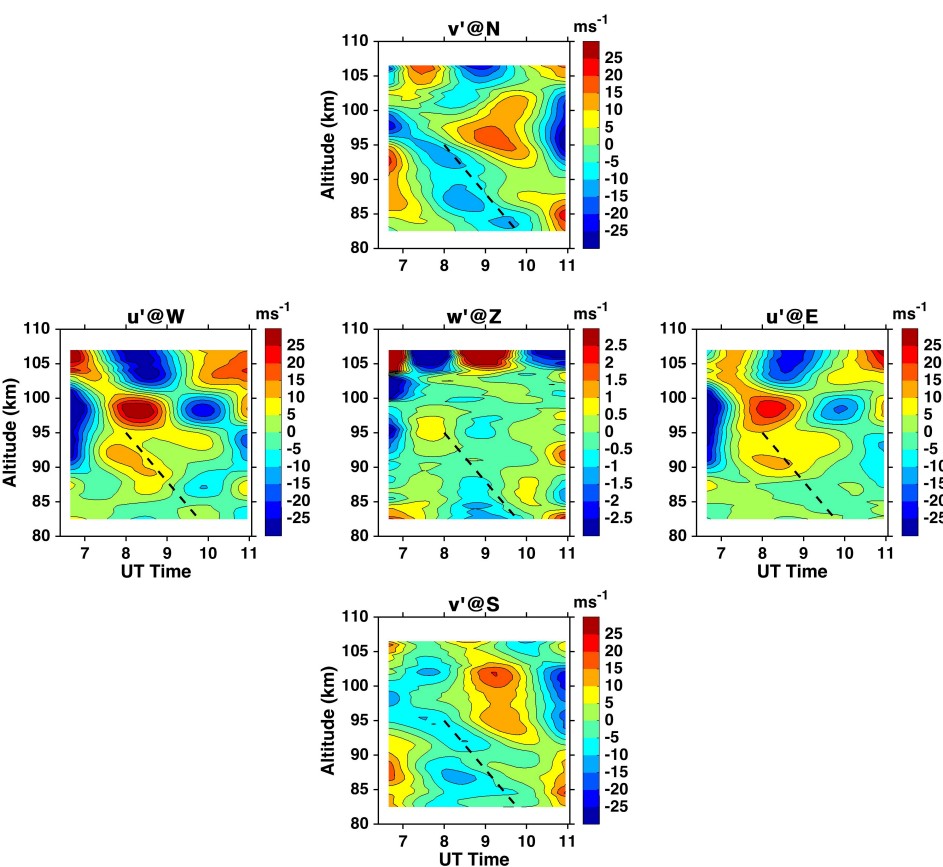

**Figure A3.** Same as Figure 6 but for horizontal and vertical winds. Note that it is zonal winds at $W$ and $E$, meridional winds at $S$ and $N$, and vertical wind at $Z$.

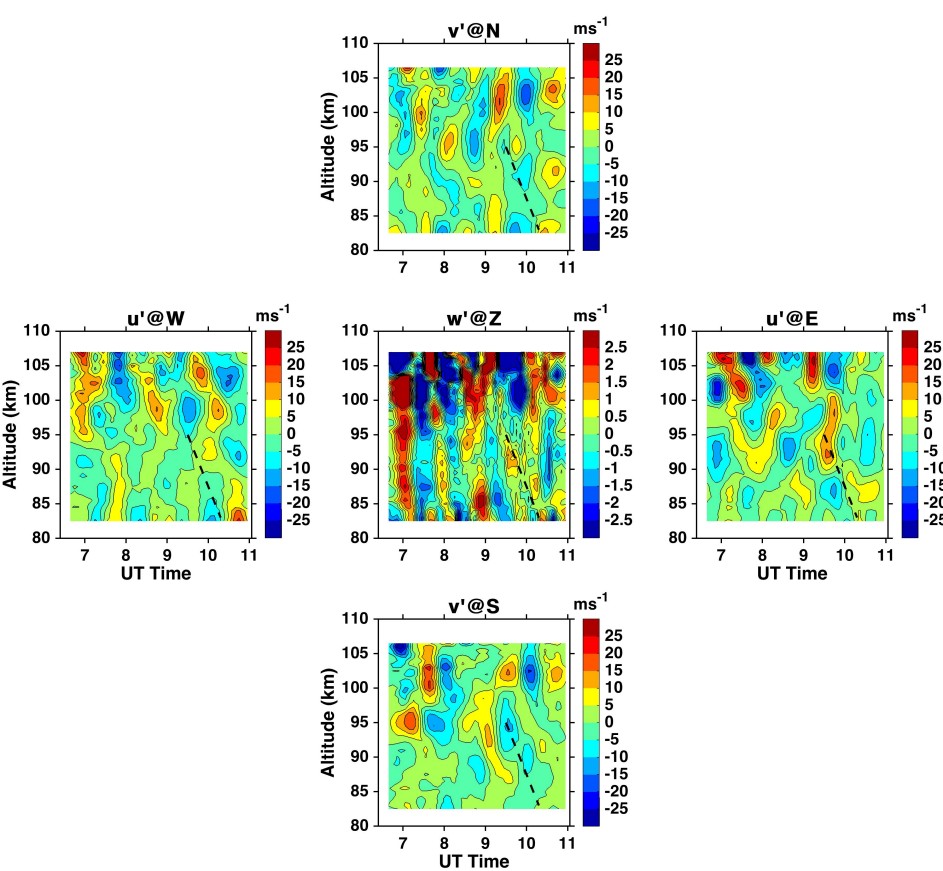

**Figure A4.** Same as Figure 7 but for horizontal and vertical winds. Note that it is zonal winds at $W$ and $E$, meridional winds at $S$ and $N$, and vertical wind at $Z$.



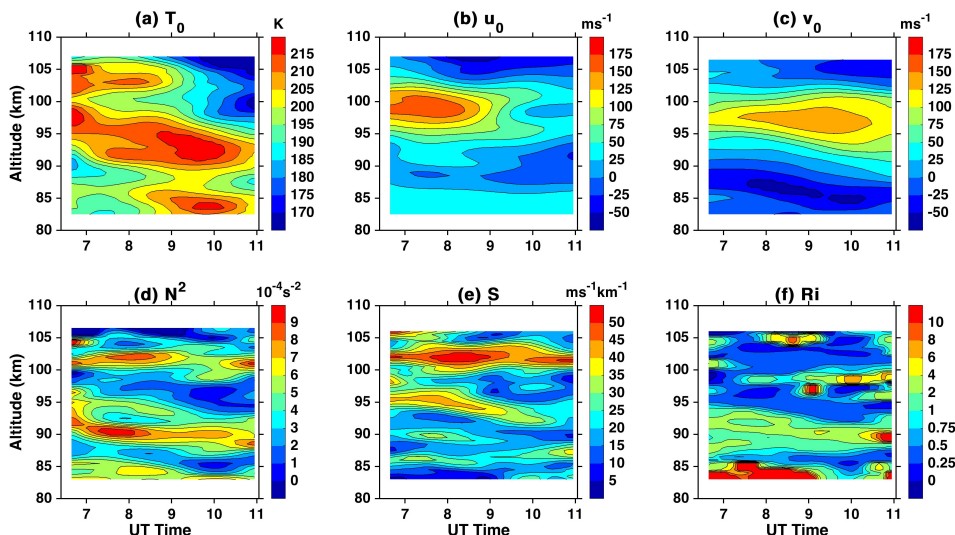

**Figure A5.** Same as Figure 5 but the background contains all perturbations with periods longer than 2.2 hour, which includes quasi-3.2 hr wave.





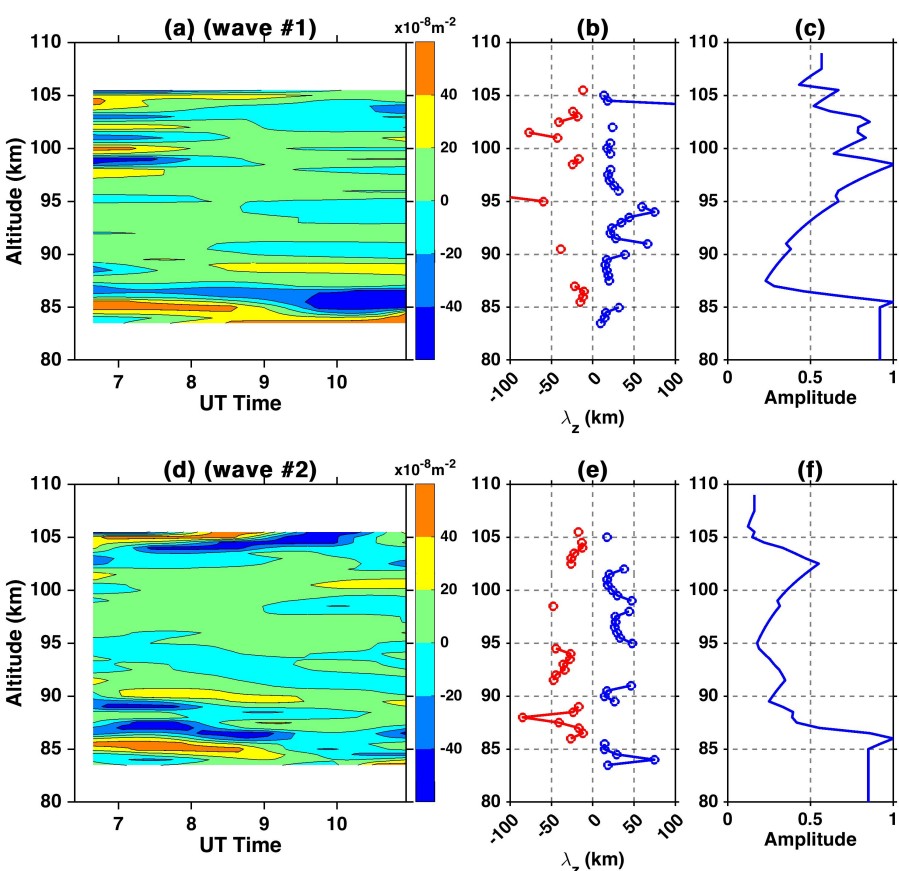

**Figure A6.** Same as Figure 9 but calculated based on dispersion relation equation (10).



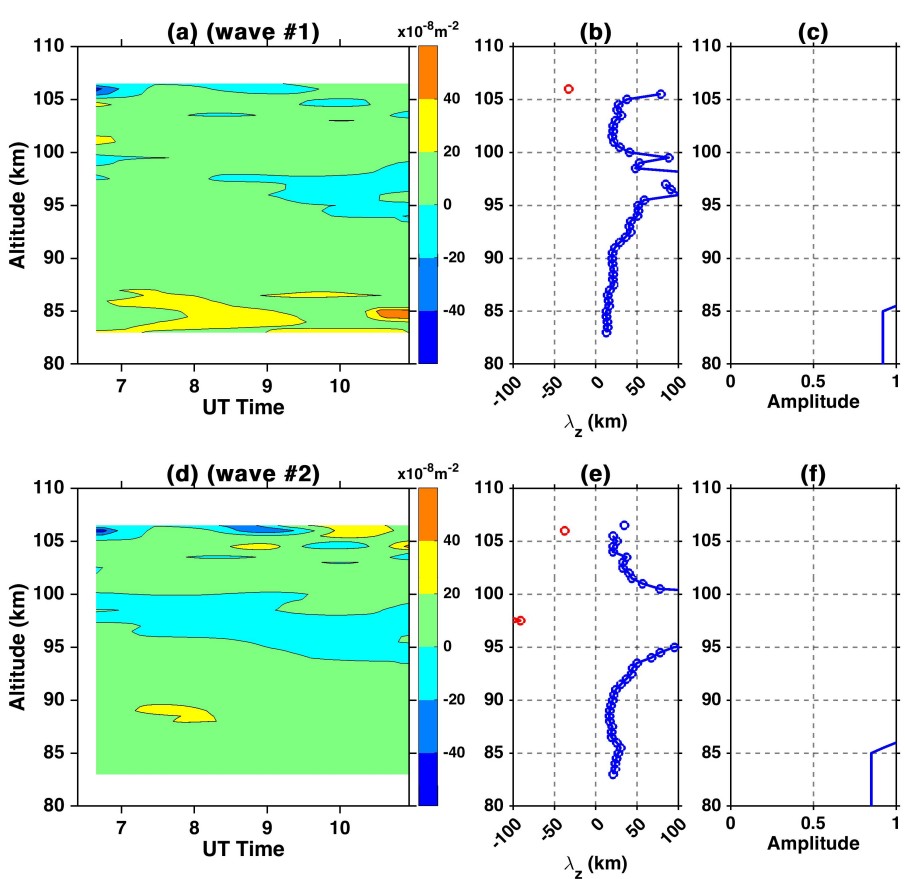

**Figure A7.** Same as Figure 9 but calculated based on dispersion relation equation (11).