# Peer review of "Investigation of Gravity Waves using Measurements from a Sodium Temperature/Wind Lidar Operated in Multi-Direction Mode"

_EGUsphere, 2023_

## Author Response (AR1)

We really appreciate the RC1 reviewer's comments and suggestions. The responses for each comment and question are provided here, highlighted in bold font, and corresponding revisions in the manuscript are also explicitly described.

In "Investigation of Gravity Waves using Measurements from a Sodium Temperature/Wind Lidar Operated in Multi-Direction Mode," the authors suggest a new method using multi-direction temperature and wind results from sodium lidar to analyze gravity waves. A case study at Maui, Hawaii was used to illustrate the method's feasibility. Using lidar observation data, the authors get the parameters of the GWs, such as period, wavelength, propagation direction, and so on. The GWs' propagation and dispersion conditions can also be calculated and well-explained with wave theory. In addition, the sensitive study has been presented to state that the method is suitable for medium-scale and medium-frequency GWs. Overall, this work presents a new and reliable idea for calculating the GWs parameter with the multi-direction measurement and is suitable for publishing after minor revision.

Minor comments:

In Figure 6 and Figure 7, the sentence "Overlapped contours are the Richardson numbers with values 0.25 in blue and 0.5 in black." There are no black contours in these 2 figures.

Because the Ri=0.5 contours spread over a large area on the figure, it is originally plotted with much thinner contour lines so not to obscure the color shadowing in the background. In the revised manuscript, we updated the figure to make both the blue (Ri=0.25) and black (Ri=0.5) contours show up in the same width.

Line 309, "In Figure 9(c) and 9(f), the relative wave amplitudes are estimated, assuming a unit amplitude at the lowest altitude." What is the value of the unit amplitude?

In this context, we calculated the theoretical wave amplitude variations considering the evanescence layers (exponential decay) and energy conservation (exponential growth). We focused on the relative changes of the wave amplitude, chose a unit or 1 K wave amplitude at the bottom, and estimated the wave amplitude at higher altitudes. We revised the manuscript to make it clear that the 1 K wave amplitude is chosen as the lowest amplitude for the calculation. Both Figures 9(c) and 9(f) are updated to be more precise.

Line 393, "The retrieved phase shift shows an error of within 10 % for all periods and wavelengths shorter than 2000 km." For the light blue line shown in the bottom of Figure 10, which wavelength is 1000 km, the phase shift error can reach 20%. Please check the sentence and make sure the statement is right or not.

We agree with this point. The phase shift is much more sensitive to the errors. We corrected the statement that a typical 20 % percentage error is more proper for the phase shift for waves with wavelengths shorter than 2000 km.

We really appreciate the RC2 reviewer's comments and suggestions. We provided the following feedback and revisions as reflected in the manuscript, all highlighted in bold font.

Review of "Investigation of Gravity Waves using Measurements from a Sodium Temperature/Wind Lidar Operated in Multi-Direction Mode" by Cao and Liu, submitted to Atmospheric Measurement Techniques (egusphere-2023-1563).

In this manuscript, the authors present gravity events observed by a narrow-band sodium lidar on 14 January 2022 at Maui, Hawaii. A novel method based on cross-spectrum was applied to the lidar data to retrieve the full characteristics of the two wave packets that were identified. Propagation and dissipation of the waves were analyzed and discussed under the known background conditions as measured by the lidar. They found that t both wave packets propagate through multiple thin evanescent layers and are partially reflected but part of the energy still penetrate higher altitudes. They also used forward modelling to demonstrate the sensitivity of this novel method and concluded that it's most sensitive to medium-scale, medium-frequency waves.

In general, the manuscript is well-written. The results are presented clearly and the conclusions are reasonable. The results and conclusions are scientifically significant as lidar is an important tool to investigate gravity waves, an important phenomenon that couples the whole atmosphere system.

I do have one question and I think the manuscript is publishable after it's resolved. In the case where one have measurements in both zonal and meridional wind, another (and a very common) method to retrieve full characteristics (especially the intrinsic parameters) of GWs is the hodograph method. Given that in the lidar data, we do not know the 'real' parameters of the wave, it would be useful to compare the parameters of the waves using this novel method, and the hodograph method. Maybe no figures or anything but a comparison of the numbers will be significant.

We actually tried the hodograph method in our analysis. However, we did not get consistent and convincing results. We attribute this to the following two reasons. First, it turns out the hodograph method mostly applies to the inertia gravity waves with longer periods that are comparable to the inertia period. At Maui (20° N), the inertia period is estimated to be 33.8 hours, while the retrieved waves in this study have periods of <3 hr. A ratio larger than 10 makes the hodograph method improper for the gravity waves identified in this study. Second, our method treats the observations retrieved from different laser beams as independent, and there are noticeable phase shifts among them, at least for targeted waves presented in this study. Therefore, the zonal winds (W and E) and meridional winds (N and S) are not at the same locations. To apply the hodograph method, we tried to correct the phase shift, but it turns out the uncertainties of the data made it hard to estimate a reliable phase from the winds, and also hard to apply hodograph analysis.

Even though we advertised our method as supplementary to the hodograph method to retrieve horizontal wave information, our method mostly applies to waves of shorter periods than inertia periods.

Other editorial comments

Line 399, "As lidars usually provide measurements…" should be "As sodium resonance lidars usually provide…" since other lidar systems such as Rayleigh lidar or troposphere lidar can provide measurements in other ranges.

In this context, we focus on the narrow-band resonance lidars that could be operated in multi-direction mode. We corrected this sentence to make it specifically about narrow-band sodium lidar targeting the mesosphere region.

We really appreciated the community reviewer's efforts in reviewing our manuscript and providing many useful comments and suggestions. Here, we provided detailed responses to all the comments and questions. If any revisions are made to address them in the manuscript, they are described here as well. Multiple comments are about very similar topics, such as the methods and authenticity of the waves. We will address some key comments and concerns at the beginning and respond to other comments in later sections. Due to the size limit, all the figures are only included in the PDF attachment, where the full responses can be found, including everything listed here.

[Figure]

(1) Regarding the methodology, this study presents a new method to derive the gravity wave horizontal information from Na lidar observations. To clarify the method, we add an extra subfigure in Figure 1(b), showing the simulated time series with phase shift or difference at three different locations (East, Zenith, West). The small phase shift is due to a spatial separation of ~50 km between laser beams at Z and E/W at 90 km altitude. In summary, we tried to identify this **small phase shift (or phase differences)** from the observations (time series) in different directions. Then, we can derive the horizontal wavelength and propagation direction from the phase shifts in the E-W and N-S directions. We have detailed descriptions of the proposed methods in Section 2. The implementation of the method is mathematically simple. First, apply some interpolation, the phase shift is derived from the cross-spectrum of two time series. Then the wave parameters (horizontal wavelength, phase speed, and propagation azimuth) are calculated from the phase shift based on equations (5) and (6) in Section 2. Next, the whole process is repeated for data of all the altitudes. The whole process can be simply described as: **two time series -> interpolation -> cross-spectrum -> phase shift -> wavelength/wavenumber -> wave speed -> wave vector.**

(2) Regarding the authenticity of the derived waves, we acknowledged several times the existence of spectral leakage due to limited data samples and coarse resolution in the manuscript. We used caution in each step to make sure the wave signatures are real. Our method did not only rely on the FFT-retrieved wave amplitude/period but also depended on **the consistent phase shifts in the time series, and the phase shift must satisfy the wave propagation features**. We declared that the retrieved perturbations were actual wave signatures, not aliased signals from tide residuals or artifacts of random noise, only after we identified consistent phase shifts in the time series and derived consistent wave vectors. As the

reviewer pointed out, multiple methods were implemented in the data processing, including interpolation, filtering, and FFT. They are carried out independently on the measurements from different directions. After these procedures, the phase shift, like Figure 1(b), shows up in the time series at different altitudes, and the phase shift follows the propagation feature, like W->Z->E and N->Z->S, so we believe that they are not artifacts or random noises. Our main arguments are: (1) the atmospheric tides have very large horizontal scales and would not generate a noticeable phase shift within a 100 km distance (also see latter sensitivity analysis on wave scales). (2) the probability of random noise shown in five directions and associated with consistent phase shift, is close to zero. (3) even if the spectral peaks are not very accurate, they should be very close to the true values for wave packets, and the phase shift of the wave packet will still be prominent.

To support these arguments, we attached a figure (right below) of the time series of the 1.6-hour wave at 90 km after filtering. Just like the ones shown in the simulations in Figure 1(b), clear phase shifts can be seen in the measurements from different directions. The wavefront moves in the directions of E->Z->W and N->Z->S. The perturbation amplitudes reach about 8 K and 10 m/s in temperature and winds. Please note that the circles of different colors demonstrate the raw measurements before any interpolation is applied in the time domain. With the raw 5-min or 10-min resolution, we can resolve the phase shifts of a 1.6-hr and 3.2-hr wave.

**To recap, we have the following evidence to support the authenticity of the waves: (1) perturbations with noticeable phase shifts in the time domain (Figure 6/7), (2) consistent peaks in the spectral domain (Figure 4), and (3) consistent wave vectors over the altitude range (Figure 8b/8c).**

[Figure]

(3) Regarding the potential spectral leakage, we acknowledged the limited data samples could bring uncertainty in the spectral peaks. So, we tried our best and used caution at every step to eliminate the artifacts. First, we examined the temperature and wind perturbations in the time domain, and almost confirmed the existence of the waves and estimated the wave periods. In this case, the FFT is primarily used to determine the phase shifts. Once we determined the

phase shift and estimated the wave vectors to be consistent, we assert there are actual gravity waves being detected, not aliased tide signals or artifacts, which is shown by the consistent wave vectors in Figures 8(b) and 8(c). The derived two wave packets show different propagation directions and are consistent over certain altitude ranges, so we declare two possible waves are identified instead of two leaked signatures from the same wave. Figure 8(c) shows wave vectors (length = speed, orientation = azimuth) for the 1.6-hr wave. The wave vectors are very consistent, only below 91 km. If such a wave does not exist, the wave vector will be like those above 92 km in 8(c), showing random orientation (propagation azimuth) and length (phase speed). **In summary, it is the consistent phase shift that helped us to confirm the existence of the wave.**

(4) Regarding the novelty of this manuscript, we did not claim the calculation of the Ri, N2, and m2 from gravity wave parameters to be a big breakthrough, as they are indeed very routine methods based on linear gravity wave theory. However, we tried to advertise the **cross-spectral method of deriving phase shifts that can be used to estimate full gravity wave parameters for this type of Na lidar**. In this proposed method, we only assume the target waves are close to monochromatic waves, such as wave packets. In gravity wave observation, it is generally challenging to derive the horizontal and vertical wave parameters simultaneously unless multiple complementary instruments observe them simultaneously. In the past, people used the hodograph method to derive horizontal wave information for longer-period inertial gravity waves. This study provides a second possible method to derive horizontal wave information for medium-frequency gravity waves, which hopefully can make up some spectral gaps in observing gravity waves.

(5) Regarding the propagation error, we acknowledged that this old dataset from an old lidar system has relatively large errors, we estimated the uncertainties from the standard deviation of the results along altitudes. We determined two different gravity waves were identified from this old dataset and show all the related figures. The importance is to demonstrate the new method and we chose to present results as they are, at the same time, acknowledged the drawbacks. We hope newer lidar systems could provide better datasets to verify it.

(6) Regarding all the presented results about winds, we want to point out that this proposed method mainly relies on temperature measurements to derive all the gravity wave information. A theoretical limit prevents us from implementing the method on winds. We explained larger errors in winds and method limitations lead to less evident results in winds, and we already down-tuned all the discussions about winds. Most wind results are shown for completeness of the data. They are included in the Appendix, not meaning to be hidden.

A concerning aspect of this manuscript is that the data are "detrended" to remove the background, which is likely a large-scale wave or tide. This method lacks detailed description. However, it should be noted when a linear background is subtracted from a wave such as a tide, which is sinusoidal, the residual remains near the inflection of the tide creating a false perturbation.

Putting this aside, the methodology used to analyze the residuals (if we make the assumption these residuals are waves) is otherwise very concerning. I have provided reviews of this below in case the initial point is not enough.

Unfortunately, the improper use of multiple filters, FFTs, and detrending make it difficult to discern what is actually present in the dataset versus what is an introduced artifact or noise. The color plots are interpolated as well, which is problematic for the coarse resolution of data used.

Additionally, there is no propagation of error analysis included in the calculations. The errors associated with Ri and, N^2, and m^2 are expected to be quite large, possibly even so large that these calculated values are irrelevant.

While the equations used to calculate Ri, N^2, and wave parameters such as horizontal phase speed are correct, applying these equations to data is not a novel technique and has been published many times before. The methodology used to extract data in tables 1 and 2 is not provided or clearly justified.

Specific comments are included below:

Line 144 "The detrended temperature measurements in different directions are shown in Figure 2"

What sort of detrending method was used? Why are the original data not being shown here? Detrending in the presence of a sharp tidal structure can cause the appearance of a perturbation.

Here we used a 2$^{nd}$ order polynomial fitting to detrend data to remove the possible tide signal. We compared the results of linear fitting and 2$^{nd}$ order polynomial fitting, there are some differences in the wave amplitudes; however, the phase shifts are very close in both cases. We concluded that the much larger scale tides would not generate a noticeable phase shift at a distance of 100 km. A 2$^{nd}$ order polynomial fitting should be able to get rid of most tide components. There might still exist some tide residuals in the perturbation, they mainly change the wave amplitudes and will not influence the calculation of the phase shift, which is the key to deriving horizontal wave information.

Line 145 "Abundant wave components of various periods are identified from measurements of all directions, and distinct downward phase progression is seen in the perturbations, which implies an upward wave propagation."

Are these "abundant wave components" or noise?

Those perturbations can be found in all five different directions, and consistent phase shifts are identified from perturbations in different directions and within a certain altitude range. We believe the random noise would not generate such a clear phase shift satisfying propagation characters in different altitudes. However, we agreed it is early to declare they are waves now. To make the context logical, we revised the discussion to describe the "perturbations" only without asserting them as waves.

Line 147 "A strong peak is found at around 90 km altitude from 09:00 UT onwards"

The "peak" at 90km is not in agreement with figure 4, which shows all sorts spectral power at a range of altitudes and not necessarily a peak at 90km.

We removed in the context to remove the "strong peak" claim.

Line 148-150 "The wave patterns of the perturbations in different directions are very similar, so they are likely the same wave packets spreading a larger area and captured by the laser beams in different directions. Closely inspecting the wave pattern (crests and troughs) in different directions, some shifts in time could be noticed, which are the results of the spatial separation of laser beams in different directions."

What sort of scientific interpretation was used to come to this conclusion? How are you coming to the conclusion that the perturbations are "similar" and "likely the same wave packets spreading a larger area." There appears to be a jump in temperature between 9-10UT, but it is unclear what the background temperatures are and whether this is an artifact or real.

Here, we try to describe a wave propagating through the field of view of the lidar is detected by laser beams in different directions, a scenario just like Figure 1(a).

Line 150-152 "The detrended perturbations of different wind components are shown in Figure 3, similar wave patterns with a downward phase progression can be identified in zonal and meridional winds, with an amplitude of up to 20 ms−1."

This is not readily apparent. The data shown are quite noisy. Again, the readers are not shown the original data before detrending, so there may be an artifact here.

If comparing the zonal wind at W and E and the meridional wind at N and S, there are some similarities between them. We have tested different ways of detrending to remove the possible tidal signature, there should not be noticeable artifacts here. Also, the original temperature and wind measurements have a range of 50 K and 100 m/s. With a much broader color scale, it will not help to demonstrate the perturbations with a narrower range.

Line 153-54 "The wave pattern is still clear in the vertical wind perturbation, with an amplitude +/-2 ms−1. However, the downward phase progression is less evident. This is likely due to the magnitude of perturbation being equal to or less than the uncertainty of vertical winds."

This statement is problematic for multiple reasons. First, there is a wave apparent in w' provided by the authors, but with different periods than what is shown in the other plots. So, "the wave pattern is still clear" is not relevant. "The downward phase progression is less evident" implies there is a downward phase progression that can be observed, but then it is stated that it is likely the magnitude of the perturbation is equal to or less than the uncertainty of vertical winds. What is being claimed here?

We removed some confusing discussions in the context. Basically, we try to show the waves also exist in the wind perturbations, including vertical winds, but the wave patterns are not very consistent, we attribute this to the larger uncertainties and method limitation.

Line 159 "Figure 4 shows the spectra of temperature perturbation in five directions, which all show a similar pattern. The average spectrum of all five directions is shown in the upper right corner."

This is extremely misleading. The dataset shown in figure 2 has a lot of noise structure on the order of 30 minutes to a few hours without any coherent structure. Of course applying a Fourier analysis is going to show spectral power associated with periods of one to a few hours, but this does not mean it is an actual wave. Furthermore, the dataset shown is only 4.5 hours, with an interpolation of 6 minutes. In reality, data in the off-zenith direction has an effective resolution of 10.2 minutes. So while waves with a 20 minute period could technically be detected, noise can also be observed on the order of 20 minute periods or greater.

In the study, our focus is on the 1.5 – 3 hour waves. In this study, we understand the limited spectral resolution due to data length, so we do not solely rely on the spectrum to identify the waves. We also examined the wave perturbation in the time domain and confirmed the existence of the waves, such as shown in the figure below; the wave perturbations with a period of about ~1.5 hours are clear and the phase shifts are also consistent among different directions.

[Figure]

What is most misleading about Fig 4 is the interpolation in the plots. Based on the dataset used (6:30-11UT) being 4.5 hours and an interpolation of 6 minutes (so sampling frequency of 10 points/hr), the highest frequency possible should be ~.222/hr or 4.5 hours (the length of the dataset). But the plot shows 6.4hr, 3.2hr, and 1.6hr points. Is the dataset used actually longer? Was zero padding used? Also, it is very important to point out here that there are a limited number of datapoints at these low frequencies. The plots shown in Fig 4 are heavily interpolated when in reality there are just a few data points at the lower frequencies for each altitude. Also noise can be easily interpreted as a wave. There is not necessarily coherence between the perturbations at each altitude, and the FFT could also be sensitive to noise or artifacts from detrending.

Line 161 "Overall, there exist two prominent peaks; one has a period of about 3.2-hr and the other one about 1.6 hr."

No, these are just two adjacent data points in an FFT and NOT two distinct peaks. Yes, the dataset has spectral power in this frequency range which does not necessarily imply a gravity wave with a specific period.

Line 165: "Overall, there exist two prominent peaks; one has a period of about 3.2-hr and the other one about 1.6 hr."

Given the short length of the dataset and the sampling resolution, it would be extremely difficult to distinguish these two periods from each other.

Line 169 "Two spectral peaks are quite close in the frequency domain, so we used Chebyshev type II filters with flat passband and steep transition to stopband."

Yes, these are indeed quite close spectrally, and given the short length of the dataset and the coarse resolution, it would be difficult to distinguish the two peaks using a filter. One must consider the effective resolution of the dataset. Essentially, at the sampling resolution used of 6 minutes (its technically 10, but data were interpolated), and the length of the dataset being 4.5 hours, your frequency resolution is 0.222 hr^-1. The author is attempting to distinguish between 0.625hr^-1 and 0.3125 hr^-1. The frequency resolution does not permit the application of a filter that can separate these two waves.

Line 170: "To filter out the 1.6-hr wave component, the cut-off period of a high-pass filter is selected as 2.2-hr (0.46 hr−1). To separate the background state from two wave components, the cut-off period of a low-pass filter is selected to be 6-hr (0.17 hr−1)."

The sampling resolution and length of the dataset would not permit this. The dataset is not even 6 hours. Also, weren't the data already detrended?

Additionally, applying a filter to a dataset, especially a sharp cutoff filter can create ringing and edge effects, which would make waves appear in the filtered data that aren't real.

The following replies are for the several comments above regarding spectral analysis and filtering. More responses on similar topics can be found at the beginning. In our study, before we applied the FFT to the time series, we already identified the existence of the wave signature with visible phase shift. We first checked all the time series and found all possible wave perturbations and noticeable phase shifts. With about 4.5-hr data, we can see roughly two waves with about ~1.5-hr and one with ~3-hr periods. In our earlier version of the manuscript, we applied a non-linear least-square fitting to find the periods to be about 1.6 hr. **Here, the main purpose of the FFT is to estimate the phase shift using the cross-spectrum method.** In order to let these two wave signatures show up in the spectrum. We implemented a trick to zero-padding the time series to deliberately reveal the two known wave peaks. Moreover, our wave identification did not stop here; we are very cautious about the artifacts of the filtering and checked the filtered time series to identify the wave perturbation with consistent phase shifts among all five directions. **Essentially, this is not a usual application of the FFT; we do not try to use it to determine the period and amplitudes, rather than use it to determine the phase difference of waves with known periods.**

Line 172-173: "To fully understand the propagation condition of waves, the background atmosphere states were analyzed. Figures 5(a)–5(c) show the background temperature T0, zonal wind u0 and meridional wind v0 retrieved by low-pass filtering as defined above."

What was described above was incorrect use of a filter to attempt to retrieve long period waves from a short dataset. Its not clear how the background is now obtained, especially since the waves in question were retrieved using a "detrending."

This is a little confusing, we corrected the manuscript to be more precise. The low-pass filtering does not actually bring changes in the perturbations even though it runs successfully, and the low-pass filtered background is essentially the detrended perturbations.

Line 174-175: "The background atmosphere states show clear modulation of tides, as shown by a slow downward phase progression in both temperature and winds."

Yes, the background atmosphere does have tides, and filtering effects or detrending can cause artifacts in these tidal regions that can be misinterpreted as waves. It is difficult to tell without seeing the original data, and these should be shown in addition to these "background" data.

We acknowledged the existence of tide residuals; however, these residuals would not influence the phase differences among different directions with a 100-km separation due to the fact tides are many large-scale perturbations. This is demonstrated by the sensitivity analysis in Section 4.

Figure 5 and line 185: The calculation of N^2 and Ri requires the use of multiple data points and when accounting for propagation of error, these errors can be quite high. What is the error associated with the calculated Ri and N^2?

We acknowledged the larger error existing in the whole analysis. We limited our manuscript to a demonstration of a new method and toned down the scientific analysis.

Looking at plot A1, the wind error at best is 5m/s for the off zenith beams (U, V measurements), 15m/s at 100km, and 30m/s at 105km. To calculate Ri, two points are used to calculate dU/dz and two points are used to calculate dV/dz. Even with averaging, this would result in quite a large error, especially at higher altitudes.

We agree about the larger error.  This is the drawback of this very old dataset. However, we tried to squeeze some helpful information out of it.

Furthermore, parameters such as Ri and N^2 are parameters associated with the atmosphere itself including all waves and features. Generally, a GW propagating through the atmosphere itself generates regions of low Ri and N^2, not necessarily the "background state." These calculations are not only problematic due to the significant errors, but they are not relevant as they are based on a heavily filtered background that does not include all of the localized dynamics which determine Ri and N^2.

To be honest, everything we presented here is based on a linearized wave theory, this is no background and waves in the real atmosphere. It is the simplified wave theory that separates the wave and background. However, the whole atmosphere is nonlinear. The waves do not know there is a background and vice versa. Here, we perform all the analysis from the perspective of the linear wave theory.

Figure 6 and 7: As previously discussed, there are issues with the filters applied to the data here.

Line 176: "and there is a clam layer around 90km"

What does this mean?

The background winds reach minimum magnitudes, so it is relatively calm winds in these altitudes.

Line 193: "However, the long-period (3.2-hr) wave effectively acts as the background for the shorter-period (1.6-hr) wave."

Again, this dataset cannot resolve a separate 3.2 and 1.6 hr wave based on the number of samples and length of the dataset. Effectively, these two periods are two adjacent data points in the frequency domain, and one would expect a power spectral density with higher power for lower frequencies to exist in the mesosphere in general. Even zero padding will not change this,

as it is merely an interpolation in the frequency domain.  If a sharp filter is applied essentially only allowing 1 or 2 points in a spectrum of noise, then what results is a wave. This does not mean a wave is present.

[Figure]

In our wave identification method, our method focuses on the phase shift (like the ones shown in the figure above). We visually checked the time series first before we applied the interpolation and spectral analysis. The purpose of the spectrum analysis is to find out the phase shift using cross-spectral methods. So, in general, we confirm the wave first and then apply the FFT. The filtering might generate some artifacts, but it is literally impossible that the filtering of 5 independent time series can generate the wave with a similar period and consistent phase shift. Also, to be precise, the filtering is not necessary to identify the phase shift of the waves. We can calculate the cross-spectrum before applying filtering and still reveal the two waves with phase shift, aka, directly from the spectrum results shown in Figure 4.

Figure 6: The artifact that may have arisen due to filtering and tidal removal is being referred to as a 3.35 hour wave, although an entire period of the "3.35 hr wave" isn't readily visible over the 4.5 hour dataset.

We verified the 3.35-hr wave to be associated with consistent phase shift among different directions. The tide residues would not generate a noticeable phase shift. The wave amplitude of the 3.35-hr wave is relatively smaller and, therefore, not obviously visible in the unfiltered datasets.

Line 202: "After applying the desired filters on the temperature and wind perturbations, the two dominant wave components are isolated."

What filters? This analysis appears to isolate two different Fourier components associated with the background noise spectrum.

In our wave identification method, spectral analysis is part of the process, we also verified the waves in the time domain and wave vectors determined from phase shift. Filtering is used to

showcase the wave components, which is not necessary for wave identification. See the replies above and the explanations about the methods at the beginning.

Line 203-204: "Using the improved spectral peak determination method, the exact periods of the two dominant components are determined to be 3.35 hr (0.2986 hr−1) and 1.63 hr (0.6148 hr−1)."

What is the "improved spectral peak determination method." These aren't peaks. They are two adjacent points in the frequency domain for the given dataset.

They are not adjacent points, there is another spectral point in between. Also, see other explanations about the methods.

Lines 205-215: Aside from the concerning heavily filtered data, and the contour plot that has an interpolation associated with it, it is also important to note that outside one region between 9-11UT and 87-92km in a few of the plots in Fig 7, many of these perturbations are close to the noise of the temperature measurements. The perturbations are mostly just a few K, which is also the noise associated with T between 85-95km. Note that above 100km, the noise drastically increases from 5K to 15K at 105km. Likely the data above 100km in altitude are noise.

Interestingly, the plots of the filtered winds are hidden in the appendix A3 and A4 plots with a dotted line drawn on which does not appear to follow any mathematical or analytical methodology to placement other than an attempt to convince the reader that there is a phase progression that is not actually apparent in the data.

We did not intend to hide anything, the major wave analyses are based on the temperature measurements, and we mentioned the winds have larger errors, and the wave patterns are not very consistent in winds. The whole manuscript is still complete if we only discuss the temperature measurements; we choose to show the winds for the completeness of the observations. The dotted lines marked in all the figures are mainly used to indicate the phase shift in the time domain.

Line 218: "The wave signatures of both components are evident in the horizontal winds with the visible downward phase progression"

No, the downward phase progression is not readily visible.

We removed this unclear statement.

Line 218-219: "and the node structure is clear in the vertical direction with at least two maxima at different altitudes."

The vertical winds are plotted on an axis range from -2.5 to 2.5 m/s, which is below the noise of the wind measurements even in the best cases.

We mentioned the large error in the vertical wind, so they are shown only for completeness of observations and are not used in major wave analysis, especially in wave identification.

Lines 220-225: While there is discussion of waves in the wind data, the waves are not apparent from the plots provided.

We explained the larger error and neglect of the verticals are the main causes of waves not being visible in the horizontal winds. However, if you closely check the zonal winds (W/E) and meridional winds (S/N), there are some perturbations.

Line 235-240: Assuming that the parameters used for the waves are not noise or artifacts arising from detrending or improper filtering, there were clearly many assumptions that would go in to a calculation of phase speed. How were the propagation azimuth and phase speed determined?

The methods are described in Section 2. There is little assumption (except quasi-monochromatic wave or wave packet assumption) in the estimation of the wavelength and phase speed. Following a cross-spectrum, equations (5) and (6) are used to calculate the wavelength, phase speed, and azimuth. See more detailed discussions at the beginning.

Lines 240-245: "The horizontal wavelength wave #1 and wave #2 are estimated to be around 975 km and 438 km, both with a ~20%uncertainty. The propagation azimuth angles are estimated to be 299◦ and 233◦ for two waves, both with a 15◦–20◦ uncertainties. These wavelengths and azimuths correspond to phase shifts of -32◦ and 18◦ between measurements of E-W and N-S for wave #1, and phase shifts of -65◦ and -49◦ for wave #2."

Again, there are no details on how these calculations were made, what the assumptions were, and how the uncertainty was determined.

Tab1e 1: No information is given on how these parameters were calculated.

The methods for the calculation of these wave parameters are provided in Section 2, and a forward simulation study using the method is presented in Section 4.

Line 260: "When the atmosphere is treated as incompressible and background temperature varies slowly within the vertical wavelength of the wave, we have cs →∞ and dHs/dz →0."

This is not a valid assumption, the speed of sound does not go to infinity (although this was mentioned 20 years ago in the Fritts and Alexander 2003 paper).

We follow Nappo (2012) to discuss the wave speed being much larger than the motion of gravity and sound to make the simplification effective because we focus on the gravity waves and eliminate the acoustic waves by setting cs as infinity.

Section 3.3 Wave Diagnosis:

The math here is not new, and has been used in observational analysis in many other publications. The calculations presented here use GW parameters that were determined in the previous section of the manuscript. As previously mentioned in this review, it is not clear that these are real waves. What is also problematic here is the calculation $m^2$. There is no noise calculation or propagation of error, which is presumably very high. The value of u is used multiple times in the chosen dispersion relation, each time with an associated error.

Furthermore, there is mention of "layered structures for both waves, potentially creating ducts for the gravity waves." Based on Figure 9, these layers are very small, just a few km in altitude, so it seems strange that these would create a duct for GWs that, according to Tab1e 1 have vertical wavelengths >20km.

This is more like a speculation, the vertical wavelength is a rough estimate, and we are very loose about this conclusion. In the manuscript, we removed the uncertain discussions about vertical wavelength since we focus on horizontal wave information.

More derived parameters are provided in Table 2, and again, there is no discussion of how these parameters are obtained. Despite u' and v' not being readily apparent in the provided plots, these are somehow included in a calculation necessary for the parameters presented in Table 2.

In the calculations of the numbers in Table 2, only the wave amplitude from u' and v' are roughly estimated from the filtered perturbations.  This study is mainly based on temperature measurements; most wind results are shown for the completeness of the data. We explained larger errors in winds lead to less evident results, and we down tuned all the discussions about winds.

---

## Author Response (AR3)

In the following, the responses are in blue and the corresponding revisions in the manuscript are highlighted in yellow.

Line 218-222 "Even though there are wave patterns at higher altitudes above 100 km, the wave pattern is less consistent in different directions and shows up with varying periods. Multiple thin unstable layers exist in this altitude range, so upward propagation waves might undergo nonlinear wave mean flow interaction, resulting in wave dissipation. This is also shown by the spectra in Figure 4, where the broader spectra at the higher altitudes indicate the dispersion of wave packets."
As shown in Figure A1, above 100 km and below 84 km, the uncertainties of the temperature and winds are too large, so the frequency spectra and wave patterns may mainly due to the uncertainty of the measurement.

We agree that the large uncertainties near the top and bottom sides of the altitude range are large (~10 K), which is comparable to the potential wave amplitudes. So, the resulting spectrum containing such uncertainties may not show the actual wave characteristics in these two areas. So we deleted the sentence about "wave dispersion seen in the spectrum".

Line 228-230 "The measurement uncertainties of 5–10 m s−1 are too large compared to the wave amplitudes of 10–20 m s−1 , making the slight phase shift hard to be distinguished. In later analysis, only the temperature measurements are used for the cross-spectral method to estimate wave parameters."
Note that the temperature uncertainties are larger than 5 K above 96 km and below 84 km, while the amplitudes of the two identified waves are about 10 K as shown in Figures 6 and 7, is it more suitable than wind measurements in estimating wave parameters?

I may not fully understand the reviewer's concern here. We indeed used temperature, rather than winds, to estimate wave parameters. If the reviewer means the opposite, I have the following responses. We choose to use temperature, instead of winds, to estimate the wave parameters due to multiple reasons. The main reason is that in the decomposition line-of-sight winds to obtain horizontal winds (u and v), we have to assume homogeneity **(aka, assuming no phase shift)** to separate vertical wind components. This adds additional uncertainties in the horizontal winds (u and v) if we use them to derive wave parameters **considering phase shift**. We did a simple sensitivity analysis showing that the wave amplitude in winds should be much larger than uncertainties in winds to make this method reliable. Another benefit of temperature over winds is there are measurements at the Zenith direction with higher resolution. This helps a lot to verify the phase shift and determine the wave period.

Line 353-354 ". In Table 2, the observational results are estimated with larger uncertainties, and all show discrepancies with the predicted ones" which are the observational results, which are the predicted ones in Table 2? Also in Table 2, the A(u)/A(v) and phase difference between u and

v are equal in the processing of wave propagation and wave evanescent, why? Please correct "evenescent" to "evanescent" in Table 2.

In this work, the "predicted" amplitudes/phases are calculated by the formulas using observed wave parameters, and "observed" amplitudes are directly estimated from observed wave amplitudes; no "observed" phases were estimated.

We adjusted the 1st column in the table 2 to clarify the differences:
"Wave #1 XX" changed to "Predicted (Wave #1 XX)"
"Data" changed to "Observed"

In this context, "theoretical" is not a very proper word, as observations are also involved in the calculations. We made the following changes:
"The theoretical values of" changed to "The predicted values of".

Amplitude ratios and phase differences between u and v, are not influenced by the dissipation and evanescence of the wave in the vertical direction, at least within the linear wave assumption made in this study. The polarization relation can be simplified as u/v = k/l if we ignore the inertial frequency. In this study, the k and l are assumed to be the constants. So the derived values are the same between the free-propagation and evanescent regions for each wave. The main reason for this assumption is that the lidar field of view (100 km) covers a small part of the wave field (450-900 km wavelength). It could not directly resolve any meaningful horizontal variations within such a small area; thus, we have to assume them to be constants. The realistic waves propagate in an oblique path and should have dissipation in both horizontal and vertical directions.

The typo of "evanescent" is corrected in the table.